# The intractable challenge of evaluating cattle vaccination as a control for bovine Tuberculosis

Andrew James Kerr Conlan[1]*, Martin Vordermeier[2], Mart CM de Jong[3], James LN Wood[1]

[1]Disease Dynamics Unit, Department of Veterinary Medicine, University of Cambridge, Cambridge, United Kingdom; [2]Department of Bacteriology, Animal and Plant Health Agency, Weybridge, United Kingdom; [3]Quantitative Veterinary Epidemiology, Wageningen University, Wageningen, Netherlands

**Abstract** Vaccination of cattle against bovine Tuberculosis (bTB) has been a long-term policy objective for countries where disease continues to persist despite costly test-and-slaughter programs. The potential use of vaccination within the European Union has been linked to a need for field evaluation of any prospective vaccine and the impact of vaccination on the rate of transmission of bTB. We calculate that estimation of the direct protection of BCG could be achieved with 100 herds, but over 500 herds would be necessary to demonstrate an economic benefit for farmers whose costs are dominated by testing and associated herd restrictions. However, the low and variable attack rate in GB herds means field trials are unlikely to be able to discern any impact of vaccination on transmission. In contrast, experimental natural transmission studies could provide robust evaluation of both the efficacy and mode of action of vaccination using as few as 200 animals.

DOI: https://doi.org/10.7554/eLife.27694.001

*For correspondence:
ajkc2@cam.ac.uk

## Introduction

The use of cattle vaccination for the control of bovine Tuberculosis in Great Britain is currently prohibited under national and European law. In 2013, and before the recent referendum decision for the UK to leave the EU, the European Commission indicated that any change in legislation to allow the deployment of cattle vaccination would be dependent on carrying out field trials under European production conditions. To this end, the Commission obtained detailed recommendations on the design of suitable field trials prepared by the European Food Safety Authority (*EFSA, 2013*). A consortium was commissioned by the Department of Environment, Food and Rural Affairs (Defra) to design trials that addressed all the EFSA recommendations. To support these designs, we used within-herd transmission models with parameter distributions estimated from GB data (*Conlan et al., 2012*, *2015*) to calculate sample sizes necessary to demonstrate the likely protective benefits of vaccination. Regardless of the outcome of forthcoming negotiations for the exit of the UK from the EU, the use of cattle vaccination may require international approval to maintain trade and perhaps more importantly to ensure the economic buy-in of UK Farmers. In this paper, we demonstrate that satisfying two key EFSA recommendations have profound implications for the likely benefits and necessary scale of any field trials; these are that vaccination should be used only as a supplement to the existing test-and-slaughter policy and that field trials should demonstrate the impact of vaccination on transmission rather than just individual animal efficacy (*EFSA, 2013*). Use of vaccination as a supplement, rather than replacement, to test-and-slaughter means that a successful vaccine which reduces the overall burden and transmission of disease may, nonetheless, provide

**eLife digest** Bovine tuberculosis is an infectious disease of livestock and wildlife in many parts of the world. It also can spread to humans. In the United Kingdom (UK), infected cattle and badgers contribute to its spread. To control bovine tuberculosis, cattle are tested and infected animals are slaughtered. Badgers in areas near cattle are killed to keep their populations small and reduce the likelihood of them infecting cattle. These control strategies are very controversial. Testing and slaughtering cattle is expensive and many people object to badger culling.

Developing a vaccine that would protect cattle against bovine tuberculosis is a potential alternative approach being investigated by the UK government. But such a vaccination is currently illegal in Europe because vaccinated animals may test positive for infection, creating confusion. Tests for bovine tuberculosis exist, but these DIVA (short for "Differentiates Infected from Vaccinated Animals") tests are not yet licensed for use in the UK. The European Union (EU) said it would consider relaxing its laws against bovine tuberculosis vaccination if the UK government is able to prove a vaccine is effective on farms.

Now, Conlan et al. show that the specific field trials recommended by the EU would have to be extremely large to show a benefit of vaccination. Mathematical models were used to calculate how many cattle herds a bovine tuberculosis vaccine study would need to show that it protects cattle from infection, reduces transmission of the disease, and saves farmers money. Conlan et al. show that a study including 100 herds would be large enough to prove the vaccine protected individual animals. But a trial would have to include 500 herds to show that vaccination saves farmers money.

Because transmission of bovine tuberculosis is slow in the UK, trials on working farms are unlikely to be able to measure whether vaccination reduces the spread of the disease. Instead, Conlan et al. show that smaller, less expensive experiments in controlled settings would be able to estimate the effects of bovine tuberculosis vaccination on transmission.

These results informed the UK government decision to delay farm-based studies of a bovine tuberculosis vaccine until a DIVA test is available. If vaccination and the use of a DIVA test can be proven to be effective enough to replace test and slaughter policies it could be a huge economic boon to farmers, particularly those in lower income countries.
DOI: https://doi.org/10.7554/eLife.27694.002

only limited benefit for farmers. Our analyses suggest that field evaluation of the impact of cattle vaccination on rates of transmission is unviable in Great Britain before deployment of vaccination at scale. We propose that experimental natural transmission studies (Velthuis et al. 2007) should be prioritised in order to demonstrate the mode of action of cattle vaccination before costly, and risky, field trials.

Since the 1890s, the control of bovine Tuberculosis (bTB) in cattle has depended on the use of the tuberculin skin test to identify and remove infected animals (*Francis, 1947*). In countries and regions with no significant wildlife reservoirs, test and slaughter of tuberculin-positive animals has dramatically reduced, and in the notable case of Australia eliminated bTB from cattle populations (*More et al., 2015*). Tuberculin testing as carried out during the attestation era of the 1940s and 1950s brought bTB to the brink of elimination in Great Britain (GB) (*Ritchie, 1959*). However, the subsequent relaxation of cattle controls, with the majority of herds tested every three years by the 1970s (*Wilesmith, 1983*), was followed by a steady increase in incidence in the 1980s that triggered the progressive tightening of cattle controls that continues today. Relaxation of testing in GB coincided with the identification of a wildlife reservoir of bTB in the European badger (*Meles meles*) – a legally protected species. Culling of badgers to reduce the risk of bTB in cattle herds is a highly contentious issue politically (*Grant, 2009*), scientifically (*Godfray et al., 2013*) and in the wider arena of public opinion (*Cassidy, 2012*). In this context, development of a viable vaccination strategy for badgers and/or for cattle that could reduce the costs associated with test-and-slaughter has been a long-term priority for Great Britain (*Krebs et al., 1997*).

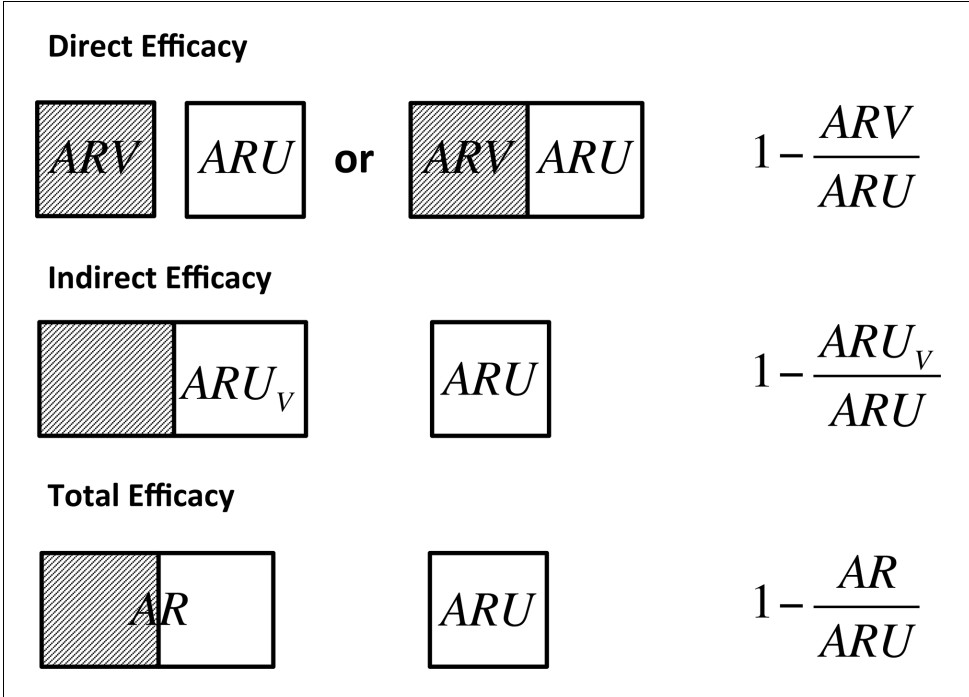

**Figure 1.** Definitions of direct, indirect and total efficacy measures. Direct efficacy can be estimated through comparison of the attack rate in vaccinated animals ($ARV$) and unvaccinated animals ($ARU$) that may be held in the same herd to control for the background infectious pressure or through comparison of fully vaccinated and unvaccinated herds. Indirect efficacy compares the attack rate in unvaccinated animals ($ARU_V$) in a partially vaccinated herd to that in unvaccinated control herds ($ARU$). Total efficacy compares the attack rate of all animals in a partially vaccinated herd ($AR$) and unvaccinated control herds ($ARU$).
DOI: https://doi.org/10.7554/eLife.27694.003

## Control of bovine tuberculosis in Great Britain

While the ultimate goal of the locally devolved strategies in Great Britain is to eliminate bovine tuberculosis from domestic cattle herds, the more economically important goal is to achieve officially TB-free (OTF) status. OTF status is defined by the EU in terms of demonstrating a long-term herd level prevalence of confirmed bTB of less than 0.1% (Council Directive 64/432/EEC). While Scotland has already achieved this goal, herd level prevalence in England and Wales continues to rise despite intensifying control measures. The current Welsh (**Welsh Government, 2012**) and English (**DEFRA, 2014**) strategies for achieving OTF status, and international trade regulations, depend on the continued use of tuberculin testing and compulsory removal of test positive animals from herds. The only viable candidate vaccine for use in cattle at this time is the *Bacillus Calmette-Guérin* (BCG) vaccine which sensitizes vaccinated animals to tuberculin and dramatically increases the likelihood of false-positive tests.

The practical and economic benefits of cattle vaccination therefore hinge on the performance of a new diagnostic test that can accurately **D**ifferentiate **I**nfected from **V**accinated **A**nimals (a so-called **DIVA** test) as much as on the efficacy of vaccination. DIVA tests for BCG have already been developed in the form of a interferon-gamma blood test (**Vordermeier et al., 2011**) and a skin test based on defined antigens (**Whelan et al., 2010**). However, both these tests must still be validated and, in the case of the skin test, approved by regulatory authorities. From the perspective of maintaining the security of international trade, and highlighted by EFSA (**EFSA, 2013**), the most important requirement for validation is that the *sensitivity* of any proposed DIVA test is at least as good as the existing tuberculin test. However, under the intensive schedule of testing in Great Britain, where affected herds are repeatedly tested until clear, it is diagnostic specificity that provides the greatest barrier to delivery of an economic benefit of vaccination (**Conlan et al., 2015**).

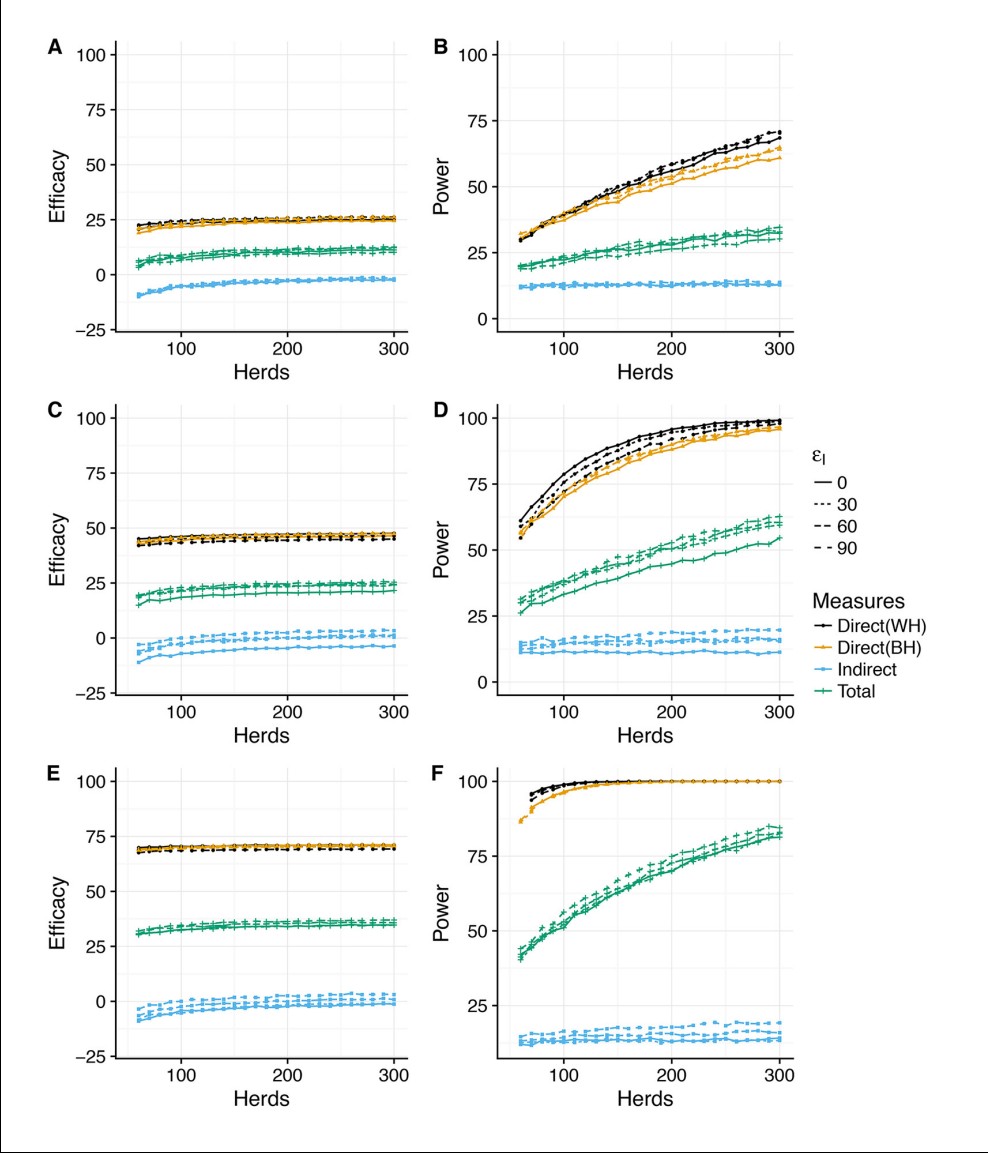

**Figure 2.** Power to estimate Direct, Indirect and Total efficacy from a two-level design (SORI model). Expected values of the Direct, Indirect and Total vaccine efficacies measured from a two-level design with trial duration of 3 years and with 75% of herds vaccinated at 50% and 25% of herds as unvaccinated whole-herd controls. We explore a range of assumed vaccine-induced reductions in susceptibility (A,B) $\varepsilon_S = 30\%$, (C,D) $\varepsilon_S = 30\%$, (E,F) $\varepsilon_S = 90\%$) and infectiousness (Linetype, $\varepsilon_I = 0, 30, 60, 90\%$). The expected effect sizes (A,C,E) are illustrated by the median of the posterior predictive distribution for each measure. Power (B,D,F) is calculated using a classical hypothesis test on the relative risk of infection (RR) in vaccinated compared to unvaccinated animals (described fully in Materials and methods section). Note that in this design Direct Efficacy can be estimated relative to either within-herd (WH) or between-herd (BH) control animals. 100 herds provides >90% power to estimate a protective direct efficacy for a true efficacy of 60%; however, >150 herds are required to achieve an 80% power to estimate a true efficacy of 30%. The indirect efficacy is predicted to be close to zero as the extrinsic force of infection acting on herds overwhelms the indirect protection provided by immunity within the herd. There is a ~50% probability of estimating a negative indirect efficacy of vaccination across the number of herds explored. As a consequence of the minimal indirect protection offered by vaccination, the Total Efficacy of vaccination with herds with 50% coverage is approximately half that of the direct efficacy. The model predicts that a 90% power of estimating a positive Total Efficacy would require >300 herds for a true direct efficacy of 60%.

DOI: https://doi.org/10.7554/eLife.27694.004

The following figure supplements are available for figure 2:

*Figure 2 continued on next page*

*Figure 2 continued*

**Figure supplement 1.** Posterior predictive distributions for Direct, Indirect and Total efficacy from two-level design (SORI model).
DOI: https://doi.org/10.7554/eLife.27694.005

**Figure supplement 2.** Power to estimate Direct, Indirect and Total efficacy from two-level design (SOR model).
DOI: https://doi.org/10.7554/eLife.27694.006

**Figure supplement 3.** Posterior predictive distributions for Direct, Indirect and Total efficacy from two-level design (SOR model).
DOI: https://doi.org/10.7554/eLife.27694.007

## Modelling the impact of cattle vaccination

To explore this issue of DIVA specificity, and the more general costs and benefits of cattle-based control measures, we previously developed and fitted dynamic herd-level transmission models that mimic the sequence of testing in GB herds (*Conlan et al., 2012*, *2015*). We compared two basic models for bTB transmission, which are distinguished by different assumed relationships between epidemiological and diagnostic latency and described fully in Appendix 1. For our purposes in this study, the SOR (susceptible, occult, reactive) and SORI (susceptible, occult, reactive and infectious) models can be considered as plausible upper and lower bounds on the transmission potential of *Mycobacterium bovis* in Great Britain.

Such dynamic transmission models are essential to predict the effectiveness of vaccination within populations due to the indirect benefits of vaccination on transmission (*Anderson and May 1992*). When some individuals in the population are directly protected from infection by vaccination (all-or nothing vaccine effect), they can no longer contribute to transmission; this leads to a further, indirect reduction in the potential spread of a disease within the population.

In order for a vaccine to be useful, it does not necessarily have to provide sterilising immunity to infection (*Smith et al., 1984*). 'Leaky' vaccines that reduce, but do not eliminate, the risk of infection of vaccinates can still control the spread of disease, particularly if the vaccine also reduces the infectiousness of vaccinated individuals. This distinction between the direct and indirect modes of action of vaccination is particularly relevant for BCG. Evidence from challenge (*Hope et al., 2005*) and natural transmission studies (*Ameni et al., 2010*) argues more strongly for a reduction in the rate of progression, with a larger proportion of vaccinated animals demonstrating a reduction in the extent of lesions than presenting with sterilizing immunity. For this reason, EFSA specified that field trial designs for the evaluation of BCG in cattle should be able to directly estimate the impact of vaccination on transmission (*EFSA, 2013*).

Experimental transmission studies can be designed such that the impact of vaccination on transmission can be directly estimated, but achieving this in the field and within an ongoing test-and-slaughter program is considerably more challenging. The UK bTB control program is complex and dynamic, with the scheduling and interpretation of tests linked to the (apparent) burden of infection within herds (*Conlan et al., 2012*). Furthermore, the removal of test-positive animals from herds as soon as they are disclosed means that the force of infection, which drives statistical power, is dependent on unobserved infection within cattle, wildlife and the wider environment. Exact likelihood-based methods of inference which can deal with this missing information, such as data-augmented MCMC (*Jewell et al., 2009*) have so far proven to be computationally intractable for bTB. As a result, published estimates of transmission rates in Great Brita have all used approximate methods of inference, depending on aggregating data at the population level from large numbers of herds (*Conlan et al., 2012*; *O'Hare et al., 2014*; *Brooks-Pollock et al., 2014*). Given the scale of data required for these advanced methods and the need for results of any field trial to be transparent and easily communicated to stakeholders, we consider them inappropriate as a framework to design field trials. Instead, we focus on the use of classical relative risk measures of vaccine efficacy (*Smith et al., 1984*; *Halloran et al., 1991*), commonly used in field trial design for human vaccines, to quantify the likely impact of BCG vaccination on transmission.

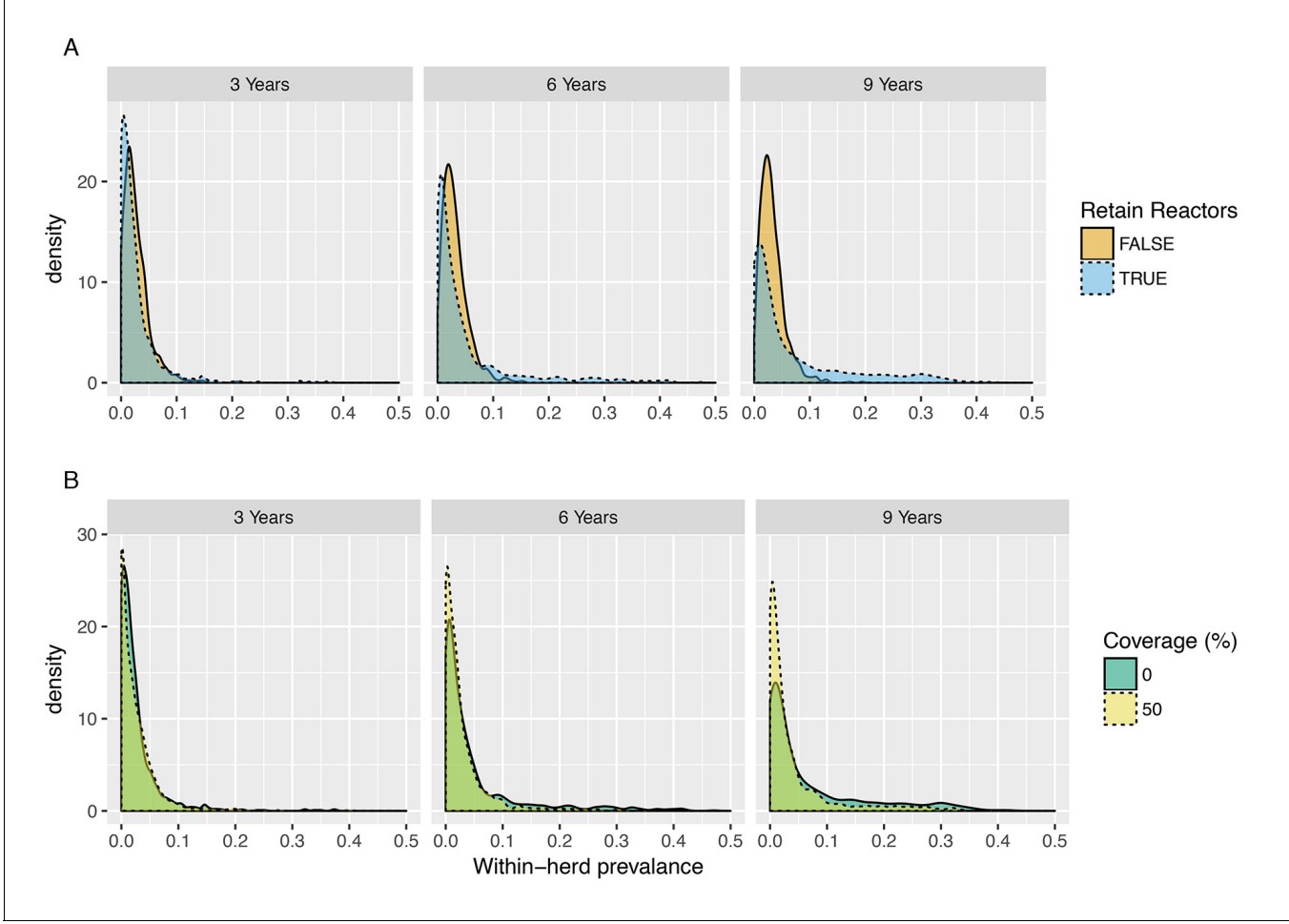

**Figure 3.** Posterior predictive distributions for within-herd prevalence for 3-, 6- and 9-year trial durations (SORI model). (**A**) Posterior predictive distributions for the proportion of reactors disclosed within a herd (within-herd prevalence) for trial-durations of 3, 6 and 9 years under policies where reactor animals are retained within herds (orange, FALSE) or removed when disclosed (blue, TRUE). Over a 3-year period, retention of reactors makes very little difference to the overall attack rate due to the relatively low transmission rates and long inter-generational period. (**B**) Posterior predictive distributions for the proportion of reactors disclosed within a herd (within-herd prevalence) for trial-durations of 3, 6 and 9 years in unvaccinated (0% coverage, Blue) and partially vaccinated (50% coverage, Yellow) herds for the most optimistic scenario of an assumed reduction in susceptibility and infectiousness of $\varepsilon_S, \varepsilon_I = 90\%$. Even for this most optimistic scenario, a 50% coverage of vaccination is predicted to have very little impact on within-herd transmission for trial-durations of up to 9 years.

DOI: https://doi.org/10.7554/eLife.27694.008

The following figure supplement is available for figure 3:

**Figure supplement 1.** Posterior predictive distributions for within-herd prevalence for 3-, 6- and 9-year trial durations (SOR model).

DOI: https://doi.org/10.7554/eLife.27694.009

## Relative risk measures of vaccine efficacy

The basic requirement to estimate the indirect benefit of vaccination from either field (*Halloran et al., 1991*) or experimental trial designs (Velthuis et al. 2007) is the inclusion of at least two groups with differing levels of vaccine coverage. By comparing the relative risk of transmission for unvaccinated individuals within herds that contain different proportions of vaccinated animals, the reduction in infectiousness of vaccinates that subsequently become infected can be estimated. For such designs, three separate vaccine efficacy measures can be defined (*Figure 1*). *Direct efficacy* quantifies the protection of individuals from infection and compares the risk of infection of vaccinated animals relative to unvaccinated animals either within the same herd or a control herd. *Indirect efficacy* compares the risk of infection of unvaccinated animals within a partially vaccinated herd to

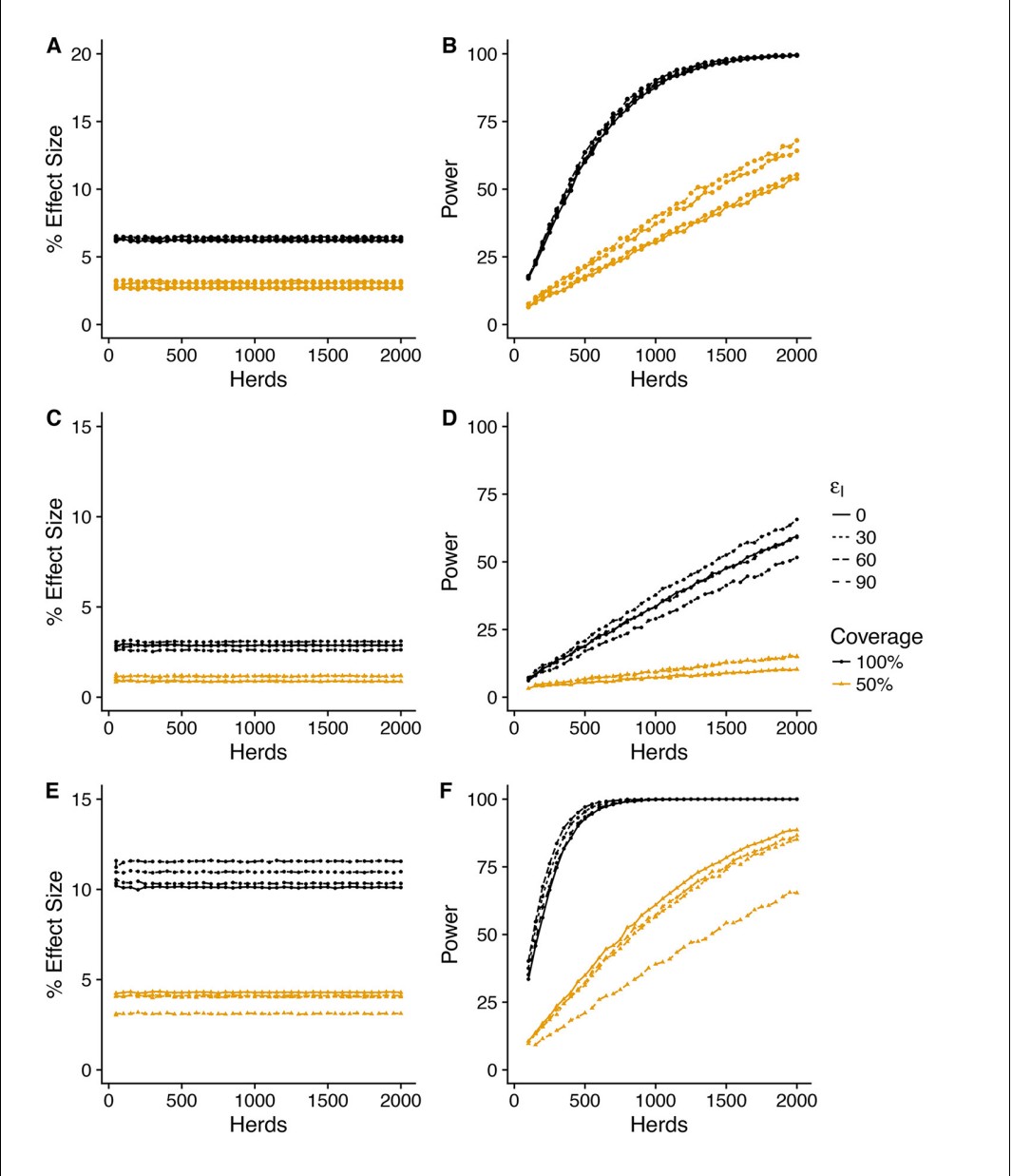

**Figure 4.** Power to estimate effect of vaccination on herd level incidence (SORI model). Simplest design to measure vaccine effectiveness with 50:50 mix of control and vaccinated herds and a target coverage of either 50% or 100% for a 3-year trial period. We explore a range of assumed vaccine-induced reductions in susceptibility (**A,B**) $\varepsilon_S = 30\%$(**C,D**) $\varepsilon_S = 60\%$(**E,F**) $\varepsilon_S = 90\%$ and infectiousness ($\varepsilon_I$ =0, 30, 60, 90%). For an assumed vaccine efficacy of 90% (reduction in susceptibility), the SORI model predicts a ~ 10% reduction in herd level incidence (defined as the proportion of herds with at least one DIVA test positive animal or slaughterhouse case) for 100% vaccination coverage and ~ 5% for 50%. For this effect size ~ 500 herds would be required to achieve the target 80% trial power for whole herd vaccination. In excess of 2000 herds (the upper range considered) would be required for a 50% within herd target coverage.

DOI: https://doi.org/10.7554/eLife.27694.010

The following figure supplements are available for figure 4:

**Figure supplement 1.** Posterior predictive distribution for effect size of vaccination on herd level incidence (SORI model).

DOI: https://doi.org/10.7554/eLife.27694.011

**Figure supplement 2.** Predicted effect of vaccination on herd level incidence (SOR model).

DOI: https://doi.org/10.7554/eLife.27694.012

*Figure 4 continued on next page*

*Figure 4 continued*

**Figure supplement 3.** Posterior predictive distribution for effect size of vaccination on herd level incidence (SOR model).
DOI: https://doi.org/10.7554/eLife.27694.013

the risk of unvaccinated animals in an unvaccinated control herd. Finally, *Total Efficacy* compares the risk of infection of all animals within a partially vaccinated herd to that in unvaccinated control herds.

We define the end point for calculation of these risk ratios as evidence of visible lesions or culture confirmation of all animals that are removed over the course of a trial due to a positive test reaction or natural turnover. Following EFSA recommendations, this controls for the impact that the imperfect specificity of bTB diagnostic tests may have on estimates of vaccine efficacy (*EFSA, 2013*).

## Herd level measures of vaccine effectiveness in field trials

Of equal importance to the efficacy of the vaccination, and essential to quantify the potential costs-and-benefits of a cattle vaccination program, is the population level effectiveness of vaccination within the existing surveillance system. The herd level effectiveness of vaccination strategies can be assessed by comparing vaccinated, or partially vaccinated, herds to whole herd controls. As the lion's share of costs associated with bovine tuberculosis, for both farmers and government, are incurred from testing and compensation, we choose to measure effectiveness of vaccination through statistical measures of within-herd persistence. Specifically, we consider the risk of breakdown (herd level incidence), duration of breakdowns and the probability of recurrence. Note that despite similar definitions, these measures are not directly comparable to published estimates of within-herd persistence in Great Britain (*Karolemeas et al., 2010*; *Karolemeas et al., 2011*; *Conlan et al., 2012*) due to differences in the scheduling of testing during the proposed trials and the replacement of tuberculin with DIVA testing for both vaccinated and unvaccinated herds (see Appendix 1).

We define the *herd level incidence* as the proportion of study herds that have a breakdown over the fixed time horizon of the trial design (3 years); prolonged breakdowns as the proportion of herds that require more than 1 DIVA test in addition to the disclosing test to clear restrictions and recurrence as the proportion of herds that experience a breakdown and subsequently see a second incident with the time horizon of the trial.

## Conceptual design to estimate vaccine efficacy and herd level effectiveness

As previously discussed, estimating the indirect efficacy of vaccination requires at least two groups with different levels of vaccination coverage. By selecting one of these groups to be a set of unvaccinated controls, a two-phase design can also be used to estimate the herd level effectiveness of vaccination. We use our herd-level simulation models to predicted the effectiveness of vaccination and calculate appropriate sample sizes for different measures of vaccine efficacy. Throughout, we aim for an 80% statistical power, defined as the probability of failing to detect a given effect at the 97.5% significance level. Vaccinated and control animals are tested at 60-day intervals throughout the trial, with slaughter of DIVA test positive animals.

Within the vaccinated group, the statistical power to estimate the direct efficacy of vaccination depends on our ability to estimate the attack rate in both the vaccinated and unvaccinated sub-populations. As such, for within-herd controls a balanced design where the target coverage is 50% will be optimal. This balanced design will also be optimal for estimating indirect efficacy for a vaccine that halves the rate of transmission. Estimates of the indirect efficacy depend on comparing the attack rate in the unvaccinated controls within the partially vaccinated group, to that in unvaccinated herds. In the case of indirect efficacy, we must balance our ability to estimate the attack rate in the within-herd controls against the effect size generated by the presence of vaccinated animals within the group. *A priori*, for an efficacious vaccine, we would expect rates of transmission to be lower in the vaccinated herds and thus we weight the design to vaccinate 75% of recruited herds, retaining 25% as unvaccinated controls. This design places a greater importance on our ability to measure the direct effect of vaccination, while still allowing for the estimation of a relatively large impact of vaccination on transmission should it exist.

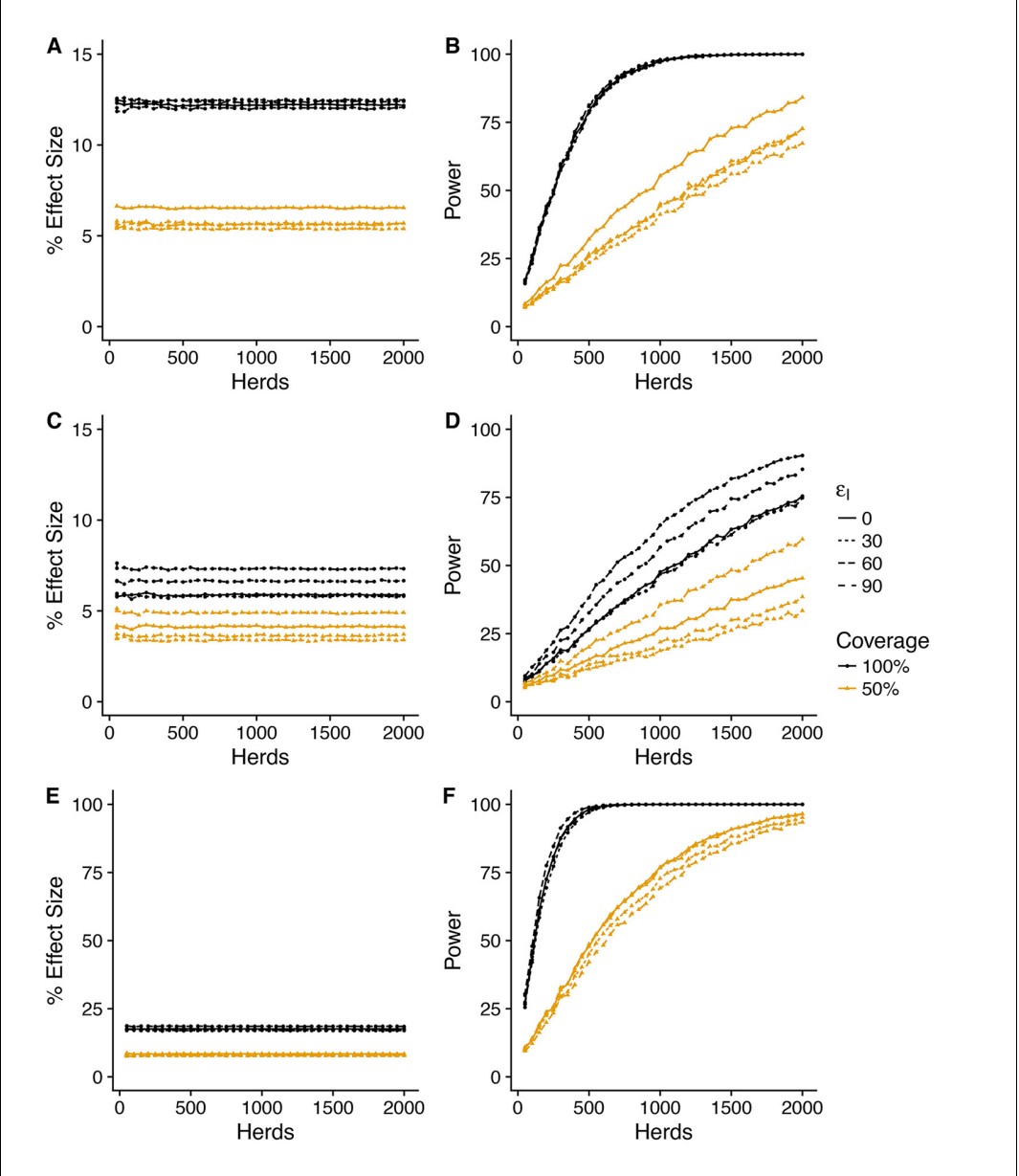

**Figure 5.** Predicted effect of vaccination on duration of restrictions (SORI model). Simplest design to measure vaccine effectiveness with 50:50 mix of control and vaccinated herds and a target coverage of either 50 or 100% for a 3-year trial period. We explore a range of assumed vaccine-induced reductions in susceptibility (A,B) $\varepsilon_S = 30\%$ (C,D) $\varepsilon_S = 90\%$ (E,F) $\varepsilon_S = 90\%$ and infectiousness ($\varepsilon_I$ = 0, 30, 60, 90%). For an assumed vaccine efficacy of 90% (reduction in susceptibility), the SORI model predicts a ~ 20% reduction in prolonged breakdowns (defined as the proportion of herds that take more than 1 short interval test to clear restrictions) for 100% vaccination coverage and ~ 10% for 50%. For this effect size, ~ 250 herds would be required to achieve the target 80% trial power for whole herd vaccination. In excess of 1500 herds (the upper range considered) would be required for a 50% within herd target coverage.

DOI: https://doi.org/10.7554/eLife.27694.014

The following figure supplements are available for figure 5:

**Figure supplement 1.** Posterior predictive distribution for effect size of vaccination on duration of restrictions (SORI model).

DOI: https://doi.org/10.7554/eLife.27694.015

**Figure supplement 2.** Predicted effect of vaccination on duration of restrictions (SOR model).

DOI: https://doi.org/10.7554/eLife.27694.016

*Figure 5 continued on next page*

*Figure 5 continued*

**Figure supplement 3.** Posterior predictive distribution for effect size of vaccination on duration of restrictions (SOR model).

DOI: https://doi.org/10.7554/eLife.27694.017

## Results

The predicted effect size of vaccination and statistical power, at least for direct efficacy, are largely consistent between our two alternative transmission models (SOR and SORI). Important differences are manifest for the indirect impacts of vaccination so both sets of model results are presented, with SORI results presented first and SOR results as supplementary figures.

Across all the considered scenarios, relative risk measures of vaccine efficacy are systematically lower than the true assumed reduction in susceptible and infectiousness. For example, reductions in susceptibility of $\varepsilon_S$ = 30, 60, and 90% correspond to predicted Direct Efficacies of ~25, 50 and 75% (*Figure 2*). This discrepancy between the true efficacy and that measured by relative risk is the consequence of the (assumed) limited duration (average of one year) of immunity (*Shim and Galvani, 2012*) and systematic biases in the relative risk measures arising from the heterogeneity in attack rate between herds (discussed further below).

The statistical power associated with these predicted effect sizes also depends critically on the variability of the posterior predictive distributions (PPD) – which for some measures is extreme. To allow for comparison between different measures, we summarise the effect size as the median value of the PPD (*Figure 2*, *Figure 2—figure supplement 2*), and plot the 95% posterior predictive intervals for the most optimistic vaccination scenario ($\varepsilon_S = 90\%$, $\varepsilon_I = 90\%$) separately (*Figure 2—supplement 1*, *Figure 2—supplement 3*).

### Direct efficacy

In this conceptual design, Direct Efficacy can be estimated relative to either within-herd (WH) or between-herd (BH) control animals (*Figure 1,2*). For an assumed direct protection ($\varepsilon_S$) of 90%, and average duration of immunity of 1 year, the power calculations are relatively insensitive to this design choice and the assumed effect of vaccination on infectiousness ($\varepsilon_I$). For this baseline assumed effect size of 90% (*Figure 2E*), which corresponds to an effective efficacy (~ 60%) comparable with existing experimental and field estimates for BCG (*Hope et al., 2005*; *Ameni et al., 2010*; *Lopez-Valencia et al., 2010*), 100 randomly selected herds in GB would comfortably provide > 90% power to estimate a positive direct efficacy for both the alternative SORI (*Figure 2E,F*) and SOR models (*Figure 2—figure supplement 1 E,F*).

This lack of sensitivity of statistical power to the choice of controls extends to lower levels of protection ($\varepsilon_S = 30\%$). However, in this scenario there is an increased sensitivity to the effect of vaccination on infectiousness ($\varepsilon_I$) and > 300 herds would be necessary to achieve the target of 80% power (*Figure 2A,B*). Alternative designs with a single target level of vaccination (distributed between or within-herds) can mitigate this reduction in power and achieve the same statistical power with 100 herds (results not shown, *Triveritas, 2014*). The necessity for designs to directly estimate indirect effects of vaccination therefore has a very real impact on the necessary scale of trials and the statistical power to estimate the basic individual level protection afforded by the vaccine.

### Indirect efficacy

In contrast to direct efficacy, estimates of the indirect efficacy are more sensitive to the choice of model with an indirect efficacy of ~0 predicted by the SORI model (*Figure 2*) and a positive indirect efficacy of up to 10% from the SOR model (*Figure 2—figure supplement 2*). This is a consequence of the different assumptions, discussed in detail in Appendix 1, concerning the time from infection to infectiousness. For the SORI model, estimates of transmission rates are higher than for the SOR model; however, animals must pass through a period of latency before becoming infectious. For the SOR model, animals have lower estimated transmission rates but are immediately infectious upon infection. As a result, the SOR model is more sensitive to the impact of vaccination on infectiousness and predicts a greater indirect benefit of vaccination.

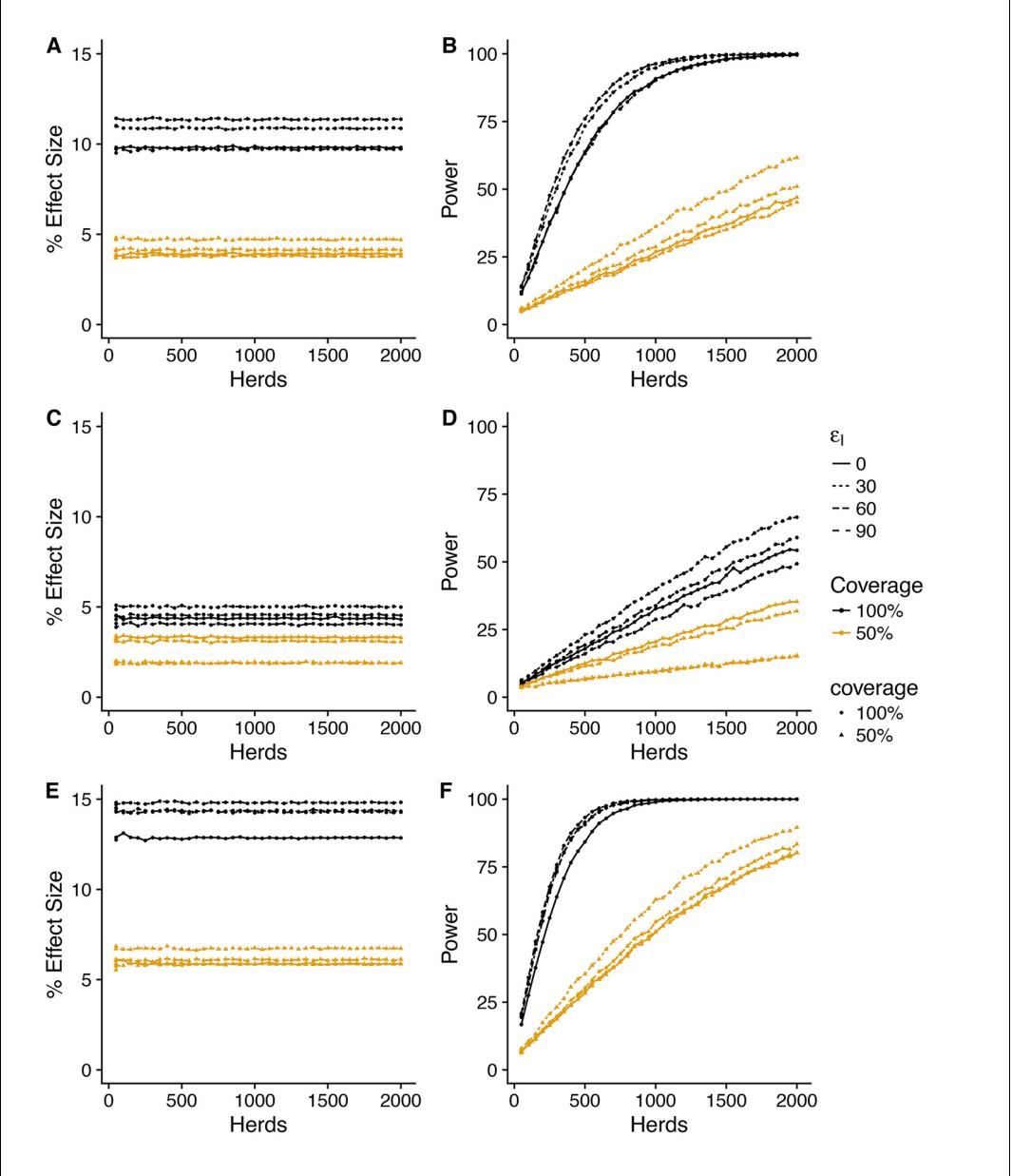

**Figure 6.** Predicted effect of vaccination on probability of recurrence (SORI model). Simplest design to measure vaccine effectiveness with 50:50 mix of control and vaccinated herds and a target coverage of either 50 or 100% for a 3-year trial period. We explore a range of assumed vaccine-induced reductions in susceptibility (**A**,**B**) $\varepsilon_S = 30\%$(**C**,**D**) $\varepsilon_S = 60\%$(**E**,**F**) $\varepsilon_S = 90\%$ and infectiousness ($\varepsilon_I = 0, 30, 60, 90\%$). For an assumed vaccine efficacy of 90% (reduction in susceptibility), the SORI model predicts a ~ 15% reduction in prolonged breakdowns (defined as the proportion of herds that take more than 1 short interval test to clear restrictions) for 100% vaccination coverage and ~ 5% for 50%. For this effect size, ~ 500 herds would be required to achieve the target 80% trial power for whole herd vaccination. In excess of 2000 herds (the upper range considered) would be required for a 50% within herd target coverage.

DOI: https://doi.org/10.7554/eLife.27694.018

The following figure supplements are available for figure 6:

**Figure supplement 1.** Posterior predictive distribution for effect size of vaccination on probability of recurrence (SORI model).

DOI: https://doi.org/10.7554/eLife.27694.019

**Figure supplement 2.** Predicted effect of vaccination on probability of recurrence (SOR model).

DOI: https://doi.org/10.7554/eLife.27694.020

*Figure 6 continued on next page*

*Figure 6 continued*

**Figure supplement 3.** Posterior predictive distribution for effect size of vaccination on probability of recurrence (SOR model).

DOI: https://doi.org/10.7554/eLife.27694.021

Nonetheless, both models predict such a small indirect efficacy that there is a high probability of estimating a negative vaccine efficacy - implying an increase in infectiousness in vaccinated animals - even when a true protective effect exists (*Figure 2—figure supplements 1*, *2* and *3*).

## Total efficacy

The magnitude of indirect protection for both models is constrained considerably by the removal of infectious animals as soon as they become DIVA positive and by the extrinsic rate of infection that captures the risk of both animal movements and the unobserved environmental reservoir. These within-herd models assume that vaccination has no impact on the reservoir of infection, hence the small magnitude of the predicted indirect benefits of vaccination. As a consequence of these two factors, the total efficacy is estimated to be approximately half that of the direct efficacy and the number of herds required to power a trial based on the total efficacy are correspondingly larger. Both models suggest that an 80% power of estimating a positive total efficacy would require >300 herds even for a true direct effect of vaccination on susceptibility of $\varepsilon_S = 90\%$ (*Figure 2*, *Figure 2—figure supplement 2*).

## Systematic bias in estimates of vaccine efficacy through relative risk measures

The underestimate of the (instantaneous) efficacy of vaccination ($\varepsilon_S$, $\varepsilon_I$) through (cumulative) relative risk measures is the natural consequence of the limited duration of immunity and dynamics of transmission within-herds. To explore how this systematic underestimate of vaccine efficacy through relative risk measures depends on trial duration and design, we examine the posterior predictive distributions for the within-herd prevalence of infection (*Figure 3*). We define within-herd prevalence as the proportion of the total at-risk population during a trial that is found to be either test-positive or culture confirmed at slaughter.

These distributions reveal a high-frequency mode of singleton (or very few) reactor TB incidents - even for trial durations extending up to 9 years. This right skewed distribution of within-herd prevalence is consistent with the distribution of reactor animals seen within UK herds where less than half of bTB breakdowns have more than one reactor animal disclosed.

The consequence of this low predicted attack rate in the majority of trial herds is a systematic underestimate of vaccine efficacy through relative risk measures. The power to discriminate between the attack rate in vaccinated and unvaccinated animal's rests almost entirely with the relatively few herds that experience a high attack rate (*Figure 3*). The origins of this variability are multi-factorial including systematic differences in the within-herd reproduction ratio resulting from the demographic structure of herds, parametric uncertainty and variability in the extrinsic (environmental) risk of infection between herds even within the same risk areas.

Herd size and the residence time of animals within a herd could in theory be used to target herds with a greater potential for transmission. However, the practicality of such targeting is limited by the relative infrequency of such herds, the necessity that participation in any trial would be voluntary and the additional requirement from the EU/EFSA that the study population for field evaluation should be representative of European production systems (*EFSA, 2013*). Targeting herds with a greater environmental risk of infection is impractical due to the lack of useful data or robust methodology to quantify these risks.

Perhaps, the most natural step to increase the risk of transmission would be to retain, rather than cull, test-positive animals for the duration of any trial. However, such action has been ruled out by policy makers due to the legal and ethical issues of leaving animals known to be infected and may pose a risk of transmission to farm workers or researchers.

Nonetheless, it is important to consider what effect this may have on the likely success of field trials. To this end, we explore the effect that retaining reactor animals has on the posterior predictive

distribution for within-herd prevalence for trial durations of 3, 6 and 9 years (*Figure 3*, *Figure 3—figure supplement 1*). We see that for a 3-year trial this, unpalatable option for policy makers, would make no difference to the predicted attack rate in unvaccinated control herds due to the relatively low cattle-to-cattle transmission rates and long generation time of bTB in cattle (*Figure 3*, panel A). Even for impractically long trials of up to 9 years, retaining reactors only serves to thicken the long tail of herds that experience a relatively high rate of transmission (*Figure 3*, panel A).

## Herd-level effectiveness of vaccination

To maintain consistency with our design for efficacy, we consider the predicted impact of vaccination for all three of our herd level measures at a target vaccination coverage of 50% and compare with an alternative design with 100% whole herd vaccination.

For all three herd level measures, the impacts of vaccination predicted by the SORI model at a baseline efficacy of $\varepsilon_S = 90\%$ (corresponding to a predicted effective direct efficacy of vaccination of $\sim 75\%$) are modest and variable, with an average improvement of between 10 and 20% for whole herd vaccination, halving to between 5 and 10% for a target coverage of 50% (*Figures 4*, *5* and *6*). As with the measures of vaccine efficacy, the predictive distributions for persistence measures manifest considerable variability with a substantial probability of observing a negative effect of vaccination even when a true protective effect exists (*Figure 4—figure supplement 1*, *Figure 5—figure supplement 1*, *Figure 6—figure supplement 1*).

As a consequence of this variation, achieving the target statistical power of 80%, would require a study population of at least 500 herds for whole herd vaccination and in excess of 2000 herds (the upper limit considered) for the target coverage of 50%. The SOR model predicts a similar, but more variable effect size (*Figure 4—figure supplement 2*, *Figure 5—figure supplement 2*, *Figure 6—figure supplement 2*), that is more sensitive to the effect of vaccination on infectiousness (*Figure 5—figure supplement 3*, *Figure 6—figure supplement 3*, *Figure 6—figure supplement 3*).

## Discussion

We have used within-herd transmission models, with parameter distributions estimated from field data in Great Britain, to calculate indicative sample sizes for field trials of cattle vaccination with BCG as a supplement to an ongoing test-and-slaughter program for bTB.

Our models suggest that evaluation of the direct protective effect of BCG in the field would be viable in the UK. A three year trial with 100 herds should provide an 80% power of estimating an individual protective efficacy of at least 30%. The scale of such a trial is affected by the requirement that test-positive animals are removed from trial herds, but is driven by the heterogeneity in within-herd prevalence of bTB in Great Britain.

At the most basic level, demonstrating the efficacy of a vaccine depends on achieving sufficient exposure of vaccinated and control animals. This is a fundamental challenge for the managed cattle herds in Great Britain where high attack rates are limited to a very small proportion of affected herds. The relatively rarity of these herds and dependence on (unmeasurable) confounding factors, such as the environmental risk of infection, makes targeting this sub-population of herds impractical and biases estimates of vaccine efficacy through relative risk ratios.

This distribution of disease has a bigger implication for the potential of field trials to measure the indirect efficacy of vaccination on transmission. For all of the conceptual trial designs and model scenarios considered in this paper, the Indirect Efficacy estimated from relative risk ratios would be essentially zero, with a high probability of estimating a negative efficacy in underpowered trials even when a considerable individual level reduction in infectiousness exists. Given the slow time-scale of bTB transmission, even the controversial step of retaining test-positive (reactor) animals within trial herds would not reduce the risk of a trial failing.

Should BCG be licensed for use in cattle, at least in the UK, vaccination will be at the discretion of individual farmers who will be expected to bear the costs of vaccination. In the UK, the major economic costs for farmers accrue with respect to the frequency of testing and the period of time under restrictions. The individual efficacy of vaccination is therefore of far less interest to farmers than the herd level effects in terms of the impact of the surveillance and testing regime on their business (*Bennett and Balcombe, 2012*).

Our models predict relatively modest improvements for farmers who would choose to vaccinate, with at most a 15% predicted reduction in the risk of a recurrent or prolonged breakdown. Part of the reason for this modest estimated effectiveness of vaccination in these model scenarios is that unvaccinated trial herds benefit from the likely benefits of the prospective DIVA test. The limited data available from challenge studies suggests that DIVA tests (*Conlan et al., 2015*) will have a higher sensitivity than tuberculin testing. The overall benefit of vaccination and DIVA testing together would be expected to be larger than the effect of vaccination or tuberculin testing alone.

Another factor likely to limit the effectiveness of vaccination in our models is the constant extrinsic rate of infection, estimated from data, that is unaffected by the level of infection within the herd. This is a pragmatic modelling assumption, taken due to the complete lack of information routinely collected on the burden of disease within environmental and wildlife reservoirs. More complex dynamic models of the reservoir could, and have been, constructed in national level models (*Brooks-Pollock et al., 2014*). However, no model can account for our basic data gap of the balance of transmission between cattle and wildlife populations that will ultimately determine the long-term outcome of vaccination (*Brooks-Pollock and Wood, 2015*). The appropriateness of our assumption of a constant reservoir will depend on the extent to which the rate of extrinsic infection into herds varies over the course of simulation. For the purposes of trial evaluation, this should be considered as a worst case scenario as vaccination will have no direct impact on reducing the infection risk within the static reservoir. However, over the (relatively) short timescale of a trial we believe this will be a reasonable approximation. The extent to which the impact of vaccination over longer time-frames will be greater depends critically on the relative rates of infection to and from the environmental reservoir (*Woodroffe et al., 2016*) and between species (*Brooks-Pollock and Wood, 2015*), the magnitude of which are highly uncertain and difficult to quantify.

A consequence of this modest predicted benefit of vaccination is that herd level effectiveness would be exceptionally difficult to estimate from partially vaccinated herds, requiring a sample size in excess of 2000 herds. This highlights once more the devastating impact including partially vaccinated herds in the design, required to estimate the indirect effect of vaccination, has on the necessary scale of trials. The number of herds required could be reduced by a three arm design which includes fully vaccinated, partially vaccinated and unvaccinated control herds. However, such a design would still require of the order of 500 fully vaccinated herds and controls, compared to 100 to evaluate the direct protection, and still have a high risk of failing to provide actionable information on the impact of vaccination on transmission.

On advice from Defra and informed by the results of this paper, the Triveritas consortium proposed an alternative to a three arm design with a phased series of trials to first evaluate vaccine efficacy, and then proceed to larger scale trials to quantify herd level effectiveness (Triveritas 2014). Such an approach to mitigating risk is implicit in the established standards for evaluation of human vaccines, a comparison that warrants further discussion.

For human vaccines, evaluation of the population level effectiveness and indirect protection of a vaccine is typically reserved for Phase IV trials, carried out after the licensing and deployment of a vaccine at scale. In this light, the EU requirement that the impact of BCG on rates of transmission should be demonstrated *before* a vaccine can be licensed is notable. Although unusual, there are important biological reasons that motivated this requirement for cattle vaccination for bTB. It is possible that the use of ineffective vaccine in combination with a less sensitive DIVA test could lead to a perverse consequence of vaccination and increase the rates of silent transmission of infection. This important question must be addressed before the widespread deployment of BCG, but we would argue that field trials are not the most effective way to achieve this.

In Appendix 2, we illustrate that an natural transmission experiment involving as few as 200 *animals* over a two-year period could provide a greater power to not only estimate the efficacy of BCG, but also the mode of action in terms of the impact on susceptibility to infection and the infectiousness. Equivalent to a Phase II trial of a human vaccine, a successful experimental transmission study could provide the confidence to go ahead with field evaluation of the efficacy and effectiveness of vaccination without the necessity to compromise the power of trials with the inclusion of partially vaccinated herds.

Our calculated sample sizes for natural transmission studies presented in Appendix 2 depend on estimates of transmission rates from field data where transmission rates scale with herd size (discussed in Appendix 3). The validity of density dependent scaling of transmission rates for small

group settings is debatable, as the empirical relationship may be the consequence of husbandry factors that correlate with herd size rather than a true dependence on group size. For this reason, we suspect that field estimates of transmission may underestimate the transmission potential in small groups allowing for a shorter contact time.

Nonetheless, the experience of previous transmission experiments with reactor animals from Great Britain (*Khatri et al., 2012*) would caution against committing to large, and potentially expensive, natural transmission study in the absence of more encouraging pilot data. Our proposed design recommends a group size of 52 animals and a contact period of 1 year in line with the more optimistic model scenarios. Endemically infected countries where the feasibility of natural transmission models has already been demonstrated (*Ameni et al., 2010*) are more promising locations for such experiments than Great Britain. However, the two-phase design of our natural transmission trial allows for a stop/go point, where phase I can be continued for an additional year if insufficient transmission is seen within the unvaccinated control animals. In this way, a trial could still provide key information on the direct efficacy of vaccination, even if low rates of transmission rule out evaluation of the indirect effects.

Experimental trials for vaccine efficacy have the advantage - and disadvantage - that extrinsic sources of infection from wildlife and the environment can be eliminated and controlled for. Such experimental designs would provide more precise information on the efficacy and mode of action of vaccination for predicting the potential impact than could realistically be achieved in a field setting. They would not satisfy the current EC requirement, and EFSA recommendation, that trials should be carried out under European production conditions (*EFSA, 2013*) or convince farmers about the practicality of cattle vaccination alongside an unmanaged wildlife reservoir. Natural transmission studies should therefore be considered as an initial screening step for any prospective vaccine before larger, more expensive and riskier trials in the field. Such field trials could (or should) be based on modelling of transmission in the cattle-wildlife system using among others parameter estimates from these transmission studies.

From challenge data, we already know that BCG has the potential to provide a protective benefit to cattle. However, our results highlight the enormous scale of trials that would be necessary to evaluate BCG alongside continuing testing in the field. The scale of such trials could be dramatically reduced by addressing the mode of action of vaccination through smaller scale natural transmission studies.

Based on our current knowledge of the likely efficacy of BCG, our models do not predict a substantial benefit of vaccination at the herd level when used as a supplement to ongoing test-and-slaughter. Indeed, the primary benefits predicted by our model come from the likely increase in diagnostic sensitivity provided by a replacement DIVA test rather than vaccination in itself. The format of the tuberculin skin test used in Great Britain – the Single Intradermal Comparative Cervical Tuberculin test (**SICCT**) prioritises diagnostic specificity over sensitivity. This is in contrast to countries who have successfully achieved TB-free status based on the use of the more sensitive Single Intradermal Test (SIT). Although not the primary focus our study, our results reinforce the benefits for management of bTB that would come from routine use of a more sensitive and equally specific test. Likewise, our results highlight that ruling out the use of vaccination as a replacement, rather than a supplement, to test-and-slaughter will inevitably limit the effectiveness and perceived benefits for farmers. Reconsidering this policy option would revolutionise the economic case for the deployment of an effective vaccine, not only in Great Britain but in developing countries which can not afford to adopt expensive test-and-slaughter programmes.

## Materials and methods

### Simulation protocols for field trial designs

For each vaccination scenario, defined by a unique level of vaccination coverage and assumed efficacy of vaccination, we simulate 5000 trials with from a sample of herds representative of the range of herd sizes and demography seen in Great Britain.

Model parameters for each simulation are sampled from approximate Bayesian posterior distributions estimated for the relevant model, as described in Appendix 1. Sensitivity to model parameters is thus implicit in our analysis, with simulations used to generate predictive posterior distributions for

the statistical measure under consideration. We use the median value of these predictive distributions to quantify the expected effect size of vaccination and the full distribution to estimate the statistical power for each measure of vaccine efficacy.

Sensitivity of our results to model structure is explored by comparing the two alternatives within herd transmission models (SOR and SORI) described in full in Appendix 1.

Simulations are initiated with no infection within herds and an extrinsic force of infection as estimated from breakdown herds in high incidence (historic annual testing) areas. Herds are initialised with no infection within the herd, and become infected at this extrinsic infection rate. Our simulated study population will therefore contain both affected (breakdown) and unaffected herds. Thus, estimated sample sizes correspond to the total number of *herds* that must be recruited rather than *breakdowns*.

As the model is fitted to breakdown herds *only,* this background rate of infection should only be considered as representative of herds with a past history of bTB. Herds with no previous history of bTB might be expected to experience a lower rate of challenge from the outside of the herd and increase the calculated sample sizes.

## Power calculations for relative risk measures of vaccine efficacy

Relative risk measures of vaccine efficacy compare the attack rate in unvaccinated and vaccinated groups within a defined population as illustrated in *Figure 1*. The attack rate within each group is calculated as the ratio of the number of cases divided by the total at risk population. For our purposes the at-risk population is defined as the total population of animals removed from herds over the duration of a trial and cases can either be culture confirmed test-positive animals or TB lesioned animals found at routine slaughter.

Direct Efficacy compares the attack rate in vaccinated animals (*ARV*) against unvaccinated control animals (*ARU*) and is calculated as:

$$1 - \frac{ARV}{ARU}$$

where ARV and ARU are calculated for each scenario using 10,000 independent samples from a pool of 5000 model simulations as described above.

Indirect Efficacy can only be measured within designs with whole herd controls and vaccinated herds with target vaccination coverage of < 100%. Indirect efficacy compares the attack rate in unvaccinated animals (*ARU_V*) within a vaccinated herd and that from unvaccinated control herds (*ARU*) and is calculated as:

$$1 - \frac{ARU_V}{ARU}$$

Total Efficacy can also only be measured within designs with whole herd controls and compares the attack rate in all animals on a partially vaccinated herd (*AR*) to that within unvaccinated control herds (*ARU*) and is calculated as:

$$1 - \frac{AR}{ARU}$$

## Power calculations for relative risk measures of vaccine efficacy for field trial designs

We base our power calculations for field trial designs upon a classical hypothesis test on the relative risk of infection (*RR*) in vaccinated compared to unvaccinated animals (Kirkwood and Sterne, 2003). We test against a null hypothesis of no difference between the two populations (*RR = 1*). To account for the high probability of estimating a negative efficacy, even when a protective efficacy exists, we use a one-sided test with alternative hypothesis *H*1: *RR* < 1. The hypothesis test takes the form of a z-test with $z = log(RR)/s.e.(log(RR))$, where the standard error of the relative risk is calculated using the standard result based upon the numbers of cases and at-risk animals in the vaccinated and unvaccinated groups. Power is then estimated based upon the empirical distribution of RR generated by sampling 10,000 independent outcomes from our pool of 5000 model simulations generated for each scenario. For each simulation, we calculate the z-statistic as described above and estimate

the proportion of simulations where z is less than the critical value (*zcr*) defining the 95% level (p=0.025 for 1-sided test). The power, defined as the probability of observing a significantly protective effect when it exists, is then calculated as the proportion of simulations where $z < zcr$.

## Power calculations for herd level measures of vaccine effectiveness

The effectiveness of vaccination at the herd level can be quantified in terms of the risk of breakdown (herd level incidence), duration of breakdowns and the probability of recurrence. Note that due to the differences in the scheduling of testing during the proposed trials these measures are not directly comparable to those previously used to quantify within-herd persistence under the current statutory regime of testing. Quantifying these herd level measures requires a design with both vaccinated and unvaccinated herds subject to the same (DIVA) testing protocol.

We consider three complementary measures of the potential effectiveness of cattle vaccination:

### Herd level incidence
The proportion of study herds that have a breakdown over the fixed time horizon of the simulation (3 years unless otherwise stated).

### Prolonged breakdowns
The proportion of herds that require more than 1 (DIVA) test in addition to the disclosing test to clear restrictions.

### Recurrent breakdowns
The proportion of breakdowns that recur within the fixed time horizon of the simulation (3 years unless otherwise stated).

All these herd level effects are all defined in terms of probabilities or proportions. We can therefore estimate statistical power for these measures using a hypothesis test on the difference between two proportions (*Kirkwood and Sterne, 2003*). We test against a null hypothesis of no difference between the two proportions (d = 0). To account for the high probability of estimating a negative efficacy, even when a protective efficacy exists, we use a one-sided test with alternative hypothesis H1: d > 0. The hypothesis test takes the form of a z-test with z = d/s.e.(log(RR)), where the standard error of the relative risk is calculated using the standard result based upon the difference d, the numbers of cases and numbers of at-risk animals in the vaccinated and unvaccinated groups. Power is then calculated based upon the empirical distribution of RR generated by sampling a given number of herds from a pool of 10,000 model simulations. For each simulation, we calculate the z-statistic as described above and estimate the proportion of simulations where z is less than the critical value (zcr) defining the 95% level (p=0.025 for 1-sided test). The power, defined as the probability of observing a significantly protective effect when it exists, is then calculated as the proportion of simulations where z < zcr.

## Additional information

### Competing interests
Andrew James Kerr Conlan: Member of a consortium led by the veterinary consultancy Triveritas that was commissioned by the Department of Environment, Food and Rural Affairs (Defra) and Welsh Government to design trials for the field evaluation of BCG for cattle in the United Kingdom (Defra Project SE 3287). Member of a Defra working group on experimental designs that led to a further project commissioned by Defra to develop the experimental designs presented in this paper that supported AJKC and MdJ (Defra project SE 32100). Currently receives research support from Takeda Pharmaceuticals for a project (on the mathematical modelling of Norovirus) that is unrelated to BCG vaccination. Martin Vordermeier, Mart CM de Jong: Member of a Defra working group on experimental designs that led to a further project commissioned by Defra to develop the experimental designs presented in this paper that supported AJKC and MdJ (Defra project SE 32100). James LN Wood: Member of a consortium led by the veterinary consultancy Triveritas that was

commissioned by the Department of Environment, Food and Rural Affairs (Defra) and Welsh Government to design trials for the field evaluation of BCG for cattle in the United Kingdom (Defra Project SE 3287).

## Funding

| Funder | Grant reference number | Author |
| --- | --- | --- |
| Department for Environment, Food and Rural Affairs | SE 3287 | Andrew James Kerr Conlan Martin Vordermeier James LN Wood |
| Department for Environment, Food and Rural Affairs | SE 32100 | Andrew James Kerr Conlan Martin Vordermeier |
| The Alborada Trust | | Andrew James Kerr Conlan James LN Wood |

The funders had no role in study design, data collection and interpretation, or the decision to submit the work for publication.

## Author contributions

Andrew James Kerr Conlan, Conceptualization, Formal analysis, Funding acquisition, Visualization, Methodology, Writing—original draft, Project administration; Martin Vordermeier, Conceptualization, Supervision, Writing—review and editing; Mart CM de Jong, Conceptualization, Formal analysis, Methodology, Writing—review and editing; James LN Wood, Conceptualization, Supervision, Funding acquisition, Project administration, Writing—review and editing

## Author ORCIDs

Andrew James Kerr Conlan (iD) http://orcid.org/0000-0002-2593-6353
Mart CM de Jong (iD) https://orcid.org/0000-0002-5339-1995
James LN Wood (iD) https://orcid.org/0000-0002-0258-3188

## Decision letter and Author response

Decision letter https://doi.org/10.7554/eLife.27694.044
Author response https://doi.org/10.7554/eLife.27694.045

# Additional files

## Supplementary files

• Transparent reporting form
DOI: https://doi.org/10.7554/eLife.27694.022

## Data availability

All data generated or analysed during this study are included in the manuscript and supporting files. Source code for all models and simulated data sets used to generate all figures is provided in a git repository.

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

## Appendix 1

DOI: https://doi.org/10.7554/eLife.27694.023

# Within-herd models for transmission of bovine Tuberculosis

To explore the likely sample sizes required for field evaluation of BCG we use individual based models previously developed to explore the potential deployment of cattle vaccination and reported in (*Conlan et al., 2015*). Briefly, these herd level models were designed to estimate the effectiveness of testing at clearing infection from herds set against empirically estimated rates of within-herd transmission and extrinsic introduction of disease from cattle-movements and environmental reservoirs. As such the complexity of the models reflects a trade-off between achieving a realistic level of biological complexity and the potential to directly infer model parameters from epidemiological data. Full details of the formulation and estimation of these models have already been published (*Conlan et al., 2015*). We describe the structure and estimated parameter distributions of these models below. Full source code for the model implementation and scripts used to generate all of the figures in this paper are available from a publicly accessible git repository: https://bitbucket.org/MonkeyMyshkin/bcgtrials. git (*Conlan, 2018*; copy archived at https://github.com/elifesciences-publications/bcgtrials).

## Modelling latency of bovine Tuberculosis

A particular challenge with modelling the transmission of bovine Tuberculosis is the uncertainty surrounding the rate of progression from infection to infectiousness and the relationship between diagnostic status and infectiousness. The traditional view of bTB progression in cattle is captured in the SORI compartmental model framework where susceptible animals (S) must progress through a series of latent classes where they are first undetectable (or occult O), detectable (or reactive to the tuberculin skin test R) before finally becoming infectious (I). However, we have found that, at the within-herd level at least, the epidemiological patterns of transmission are equally well described by a simpler SOR model where all infected animals (O,R) are potentially infectious, but transmit at a lower average rate. For the purpose of this study, these two models provide plausible upper and lower transmission scenarios as an additional sensitivity analysis to assess the robustness of the power calculations for trial designs.

The SORI and SOR models – with extensions to model the action of vaccination – are implemented as stochastic continuous time Markov processes with parameters detailed in *Appendix 1—table 1* and defined by the events and transitions in *Appendix 1—table 2*. The two models are distinguished by the reactive compartments $(R, R_V)$ being absorbing states for the SOR model with $\frac{1}{T_V} = 0$ and by the form of the force of infection $\lambda(a, t)$:

$$\lambda_{SOR} = \left( \frac{\beta(I + \varepsilon_I I_V) + \chi}{\left( \frac{H}{H_m} \right)^q} \right) RR(a)$$

$$\lambda_{SORI} = \left( \frac{\beta(O + R + \varepsilon_I (O_{V1} + O_{V2} + R_V)) + \chi}{\left( \frac{H}{H_m} \right)^q} \right) RR(a)$$

Transmission within herds non-linearly increases with the size of the herd through the (estimated) parameter $q$ and susceptibility is assumed to vary with age according to independently estimated relative risk of infection ($RR(a)$, *Appendix 1—table 3*). Herds are treated as independent, coupled to an extrinsic constant reservoir of infection $\chi$ that varies $(\chi_1, \chi_2, \chi_4)$ by the risk area that the herd is located in (the historical Parish Testing Interval (PTI) 1, 2, & 4).

## Demography of herds

The study population used to estimate these models includes a representative sample of herd sizes and management models of herds from Great Britain with a past history of bTB breakdowns. For each herd-level simulation the size of the herd is fixed, with the rate of on and off movements of animals sampled from the cattle tracing system (CTS) to generate a realistic age-structure and distribution of residence times for individual animals as described in *Conlan et al. (2015)*.

The demographic events of birth, death and movement off a herd are simulated as non-Markov processes. The time of each demographic event is sampled from an extract of the cattle tracing system (CTS) database for each individual animal included in the model. During each step of the model we simulate the time for the next Markov event ($t_{MARKOV}$) and compare to the time of the next non-Markov (demographic) events ($t_{NON\_MARKOV}$). If $t_{NON\ MARKOV} < t_{MARKOV}$, we carry out the demographic event first and recalculate a new $t_{MARKOV}$. Otherwise, a Markov (epidemiological) event is simulated and the model time updated. For a complete specification of the simulation algorithm and handling of demographic events see (*Conlan et al., 2015*).

## Model parameterisation

Distributions for key model parameters (*Appendix 1—table 1*) describing the current control regime were estimated using Approximate Bayesian Computation (ABC) based on statistical measures of herd-level incidence and persistence of bTB (*Conlan et al., 2015*). These approximate posterior distributions capture both the uncertainty in estimates of model parameters and the sensitivity of model predictions to the value of these parameters. Summary parameter estimates are presented in *Appendix 1-table 4*, full details and posterior predictive checks of these model estimates can be found in (*Conlan et al., 2015*), samples from the estimated ABC posterior distribution are provided as part of the source code for all results and figures in this paper are publicly available from the following git repository: https://bitbucket.org/MonkeyMyshkin/bcgtrials.git (*Conlan, 2018*; copy archived at https://github.com/elifesciences-publications/bcgtrials).

**Appendix 1—table 1.** Model parameters.

| Parameter | Description |
|---|---|
| $p_T$ | Standard SICCT Sensitivity. Probability of positive tuberculin test for $R, I$ individuals at standard definition. |
| $1 - p_{FP}$ | Standard SICCT Specificity. Probability of negative tuberculin test for $S, O, R, I$ individuals at standard definition (1 - probability of a false positive $p_{FP}$). |
| $p'_T$ | Severe SICCT Sensitivity. Probability of positive tuberculin test for $R, I$ individuals at severe definition. |
| $1 - p'_{FP}$ | Severe SICCT Specificity. Probability of negative tuberculin test for $S, O, R, I$ individuals at severe definition. (1 − probability of false positive $p'_{FP}$). |
| $p_{RI}$ | Slaughterhouse detection. Relative sensitivity of finding lesioned or culture positive animals ($O, O_{V1}, O_{V2}, R, R_V, I, I_V$ status) under routine inspection compared to reactor inspection. |
| $T_O$ | Occult Period. Mean length of time that animals are undetectable (occult) to SICCT test. |

*Appendix 1—table 1 continued on next page*

*Appendix 1—table 1 continued*

| Parameter | Description |
|---|---|
| $T_R$ | **Reactive Period.** Mean length of time between infection and animals becoming infectious. |
| $\beta$ | **Transmission parameter** associated with density dependence (rate per day, dimensions change with $q$). |
| $q$ | **Transmission parameter** measuring the strength of density dependence (range 0–1) |
| $\chi_1$ | **Transmission parameter** measuring infectious pressure per susceptible per year in PTI 1 |
| $\chi_2$ | **Transmission parameter** measuring infectious pressure per susceptible per year in PTI 2 |
| $\chi_4$ | **Transmission parameter** measuring infectious pressure per susceptible per year in PTI 4 |
| $H$ | **Herd size.** Sampled from empirical distribution and maintained constant through individual simulations |
| $H_M = 165$ | **Constant** equal to mid-point of range of herd sizes within study population. Used to transform density dependence of force of infection. |
| $p_D$ | **DIVA sensitivity** Probability of positive DIVA test in infected vaccinates ($O_{V1}, O_{V2}, R_V, I_V$). |
| $1 - p_{DFP}$ | **DIVA specificity** Probability of negative DIVA test for uninfected vaccinates $V_1, V_2$ (1− probability of a false positive $p_{DFP}$) |
| $\varepsilon_S$ | **Vaccine Efficacy** Reduction in risk of infection for susceptible vaccinates (status $V_1$) |
| $\varepsilon_I$ | **Vaccine Efficacy** Reduction in risk of infectiousness for infected vaccinates ($R_V, I_V$) for SORI model, and in addition ($O_{V1}, O_{V2}$) for SOR model. |
| $T_V$ | **Protective Period.** Mean length of time that animals are protected by BCG vaccination (status $V_1$) |

DOI: https://doi.org/10.7554/eLife.27694.024

**Appendix 1—table 2.** Markov events defining SOR(V) and SORI(V) stochastic transmission models.

| Event | Status | Effect | Probability per unit time |
|---|---|---|---|
| Infection | $S, V_2$ | $S \rightarrow O, V_2 \rightarrow O_{V2}$ | $\lambda(a, t)$ |
| | $V_1$ | $V_1 \rightarrow O_{V1}$ | $\varepsilon_S \lambda(a, t)$ |
| Emergence (Occult) | $O, O_{V1}, O_{V2}$ | $O \rightarrow R, O_{V1} \rightarrow R_V, O_{V2} \rightarrow R_V$ | $\frac{1}{T_O}$ |
| Emergence (Reactive) | $R, R_V$ | $R \rightarrow I, R_V \rightarrow I_V$ | $\frac{1}{T_R}$ |
| Loss of Protection | $V_1$ | $V_1 \rightarrow V_2$ | $\frac{1}{T_V}$ |

DOI: https://doi.org/10.7554/eLife.27694.025

**Appendix 1—table 3.** Relative Risk of Infection $RR(a)$.

| Age range (Years) | Relative risk |
|---|---|
| 0–1 | 1.0 |
| 1–2 | 1.8 |
| 2–3 | 2.3 |
| 3–4 | 1.5 |
| 4+ | 1.3 |

DOI: https://doi.org/10.7554/eLife.27694.026

**Appendix 1—table 4.** Summary ABC posterior estimates and 95% Credible Intervals.

| Parameter | SOR estimate | SORI estimate |
|---|---|---|
| $p_T$ | **0.50** (0.12, 0.99) | **0.51** (0.16, 0.88) |
| $p_{FP}$ | **1.7e-4** (1.5e-5, 2.9e-4) | **1.8e-4** (9.7e-6, 3.0e-4) |
| $p'_T$ | **0.8** (0.29, 1.0) | **0.79** (0.34, 0.98) |
| $p'_{FP}$ | **5.5e-4** (1.3e-4, 9.6e-4) | **5.7e-4** (1.3e-4, 9.8e-4) |
| $p_{RI}$ | **0.53** (0.15, 0.94) | **0.68** (0.16, 0.98) |
| $T_O$ | **0.18 years** (0.01, 0.34) | **0.19 years** (0.17, 0.34) |
| $T_R$ | - | **7.0 years** (2.1, 9.9) |
| $\beta$ | **1.8e-3 per year** (1.5e-4, 4e-3) | **2.5e-2 per year** (1.9e-3, 9.4e-2) |
| $q$ | **0.63** (0.32,0.94) | **0.48** (0.1, 0.9) |
| $\chi_1$ | **9.0e-6 per year** (4.0e-6, 2.3e-5) | **9.6e-6 per year** (5.1e-6, 2.2e-5) |
| $\chi_2$ | **5.0e-6 per year** (1.5e-6, 1.3e-5) | **5.7e-6 per year** (2.7e-6, 1.1e-5) |
| $\chi_4$ | **2.0e-6 per year** (5.0e-7, 8.6e-6) | **2.5e-6 per year** (7.8e-7, 5.8e-6) |

DOI: https://doi.org/10.7554/eLife.27694.027

Auxiliary parameters relating to the age-dependent risks of testing positive (**Appendix 1—table 3**), demonstrating evidence of visible lesions (**Appendix 1—table 5**) and testing positive to the SICCT test (**Appendix 1—table 6**) were independently estimated and fixed to the specified values. Parameters relating to the individual level efficacy of vaccination (**Appendix 1—table 2**) were informed by a mixture of experimental and field data as described once again in (**Conlan et al., 2015**) and subject to sensitivity analysis with respect to both the direct reduction in susceptibility ($\varepsilon_S$) due to vaccination and impact on infectiousness ($\varepsilon_I$).

## Testing and vaccination schedules for efficacy trials

To accommodate the practical requirements of field trials, herds are subjected to a simplified schedule of testing and surveillance based upon the current regulatory regime. To allow trials to be blinded, tuberculin testing is assumed to be replaced by DIVA testing for all vaccinated and unvaccinated animals on trial herds (EFSA 2013). Hence, we assume that $p_T = p_D$ and $p_{FP} = p_{FPD}$ and the test characteristics do not change with confirmation status of breakdowns.

Herds are assumed to be recruited from a high incidence area (historical annual testing interval) and have a past-history of infection but are disease free at the beginning of the trial. Simulated herds become infected at rate determined by the extrinsic infectious pressure ($\chi$). **Appendix 1-table 7** and **Appendix 1-table 8** tabulate the predicted herd level incidence for the SORI (**Appendix 1-table 7**) and SOR (**Appendix 1-table 8**) models for a three-year trial period and different combinations of vaccine coverage and efficacy.

**Appendix 1—table 5.** Age stratified probability of reactors with visible lesions/culture positive $p_{VL}(a)$.

| Age range (Days) | Probability of VL/Culture |
|---|---|
| 0–200 | 0.58 |
| 200–400 | 0.59 |
| 400–600 | 0.57 |
| 600–800 | 0.50 |
| 800–1000 | 0.41 |
| 1000–1200 | 0.31 |
| 1200–1400 | 0.29 |
| 1400–1600 | 0.27 |
| 1600–1800 | 0.26 |
| 1800–2000 | 0.26 |
| 2000–2200 | 0.28 |
| 2200–2400 | 0.28 |
| 2400–2600 | 0.27 |
| 2600–2800 | 0.28 |
| 2800–3000 | 0.32 |
| 3000–3200 | 0.27 |
| 3200–3400 | 0.31 |
| 3400–3600 | 0.28 |
| 3600–3800 | 0.27 |
| 3800–4000 | 0.28 |
| 4000–8000 | 0.28 |

DOI: https://doi.org/10.7554/eLife.27694.028

**Appendix 1—table 6.** Time-dependent sensitisation of vaccinates to SICCT (*Whelan et al., 2011*)

| Time from vaccination (Days) | SICCT interpretation | Probability classified as reactor |
|---|---|---|
| 0–90 | Standard | 0.0 |
| | Severe | 0.0 |
| 90–180 | Standard | 0.60 |
| | Severe | 0.84 |
| 180–270 | Standard | 0.80 |
| | Severe | 0.95 |
| 270–360 | Standard | 0.09 |
| | Severe | 0.30 |
| 360–450 | Standard | 0.05 |
| | Severe | 0.10 |
| 450–720 | Standard | 0.11 |
| | Severe | 0.30 |
| 720+ | Standard | 0.10 |
| | Severe | 0.30 |

DOI: https://doi.org/10.7554/eLife.27694.029

**Appendix 1—table 7.** Herd level incidence for SORI model (3 Year trial period).

| Vaccine coverage | $\varepsilon_S$ | $\varepsilon_I$ | Incidence (Herds) |
|---|---|---|---|
| 0% | 0% | 0% | 92% |
| 50% | 30% | 0% | 91% |
| 50% | 60% | 0% | 89% |
| 50% | 90% | 0% | 88% |
| 100% | 30% | 0% | 90% |
| 100% | 60% | 0% | 86% |
| 100% | 90% | 0% | 82% |

DOI: https://doi.org/10.7554/eLife.27694.030

**Appendix 1—table 8.** Herd level incidence for SOR model (3 Year trial period)

| Vaccine coverage | $\varepsilon_S$ | $\varepsilon_I$ | Incidence (Herds) |
|---|---|---|---|
| 0% | 0% | 0% | 92% |
| 50% | 30% | 0% | 91% |
| 50% | 60% | 0% | 90% |
| 50% | 90% | 0% | 89% |
| 100% | 30% | 0% | 89% |
| 100% | 60% | 0% | 87% |
| 100% | 90% | 0% | 83% |

DOI: https://doi.org/10.7554/eLife.27694.031

Herds are re-vaccinated on an annual schedule, with the entire herd vaccinated on day 0. To reduce the frequency of researcher visits to herds during trials, imports and births into the herd are assigned as vaccinates or controls on entry to a herd, batched and vaccinated at 180 day intervals and then re-vaccinated according to the herd schedule. Therefore, even for a 100% target vaccination coverage the instantaneous level of vaccination coverage will change dynamically over a simulation. The variability in coverage over time will depend on the demography of the herd and in particular the rate of demographic turnover (*i.e.* moves on/moves off). This turnover of animals will limit the effectiveness of vaccination at the herd level contrasted to that which could be achieved under more controlled experimental conditions.

Breakdowns are triggered on trial herds by the failure of routinely scheduled 6 monthly tests or by detection of a lesioned animals at slaughter. Once infection is disclosed in a trial herd it is subject to 60 day short interval DIVA testing. After a single clear DIVA test, short interval testing is suspended and follow-up 6 month and 12 month tests are scheduled. In contrast to the status-quo in GB where test-intervals are variable and subject to veterinary discretion (*Conlan et al., 2012*), trial herds are assumed to be tested at these precisely specified intervals.

DIVA test positive animals are removed from herds and subject to slaughterhouse inspection. The probability of animals having visible lesions is assumed to depend on infection status and age only. The probability of lesions depends on the empirical relationship between age and the probability of visible lesions being detected (*Appendix 1—table 5*) as estimated in (*Brooks-Pollock et al., 2013*) and previously used in (*Conlan et al., 2015*).

Given the intention of trials to inform the likely benefit of vaccination within an ongoing schedule of surveillance testing we assume that a validated DIVA test with specificity of at least 99.85% and sensitivity of at least 73.3% will be available and is a necessary requirement for progressing to any field trials of cattle vaccination. These values correspond to the break-even scenario considered in our previous modelling study where the costs of additional false

positive DIVA results balance the expected benefits of a cattle vaccine with at least 60% protective efficacy for an average duration of protection of 1 year (*Conlan et al., 2015*). Data from animal challenge studies suggests that this break-even specificity level of >99.85 is achievable in vaccinates with an interferon-gamma DIVA sensitivity (relative to visible lesions) of 73.3% (95% CI: 61.9, 82.9%) (using ESAT-6, CFP-10, Rv3615c antigens) (GJ Jones, M Vordermeier, personal communication).

## Appendix 2

DOI: https://doi.org/10.7554/eLife.27694.032

### Alternative natural transmission study design to evaluate vaccine efficacy

In this appendix we carry out sample size calculations for an alternative natural transmission study design to evaluate the mode of action of BCG vaccination. A natural transmission study has the key advantages compared to field trials that test-positive animals can be retained through the course of the experiment (*Appendix 2—figure 1*). We consider the following design:

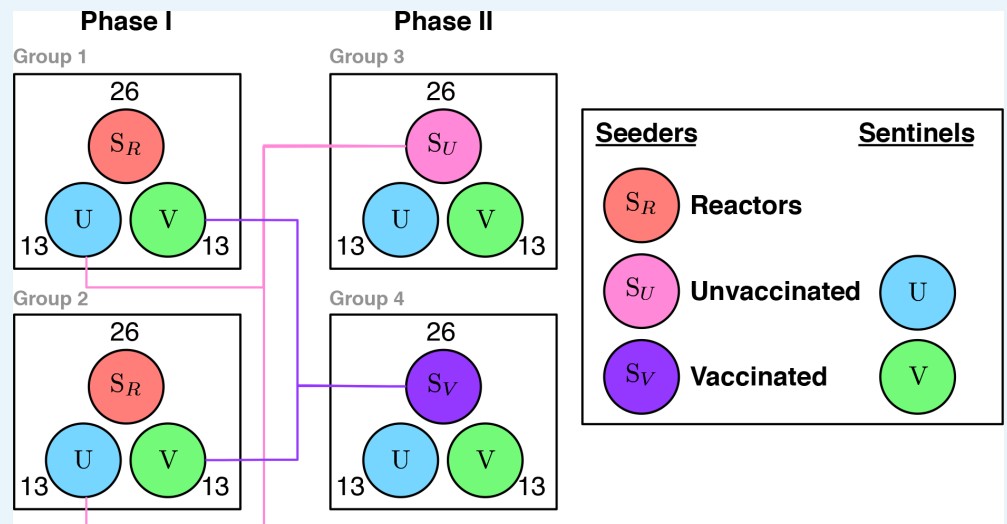

**Appendix 2—figure 1.** Experimental transmission study to demonstrate vaccine efficacy Both phases consist of two experimental groups with $52 (4N)$ co-housed animals. Within each experimental group there is a balanced number of seeder $(26, \ 2N)$ and sentinel animals $(26, 2N)$. 50% of the sentinel animals are vaccinated $(N = 13)$ and 50% unvaccinated controls $(N = 13)$. For Phase I the seeders $(S_R)$ are recruited from the field. At the end of Phase I, the exposed control animals and vaccinated animals are group together and used as the seeder animals $(S_U$ and $S_V$ respectively) and set in contact with a fresh set of (uninfected) sentinels.
DOI: https://doi.org/10.7554/eLife.27694.033

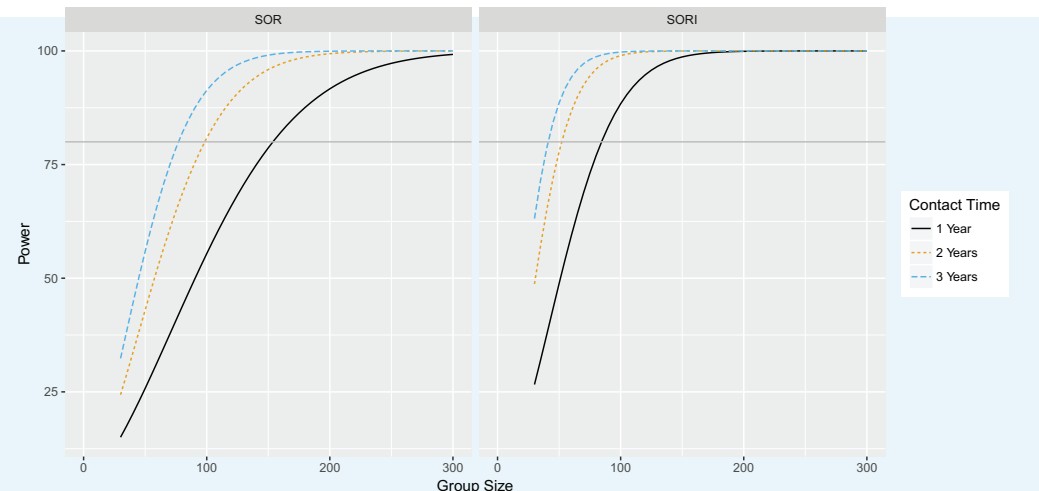

**Appendix 2—figure 2.** Power to estimate 75% total vaccine efficacy for SOR and SORI parameter estimates for contact times of one, two and three years Statistical power as a function of group size to estimate a significant difference in vaccinated and unvaccinated groups at the 95% level for an effect size of 75% total vaccine efficacy based upon SOR and SORI parameter estimates. Grey line indicates the 80% power level.

DOI: https://doi.org/10.7554/eLife.27694.034

The following figure supplements are available for figure app22:

**Appendix 2—Figure 2 supplement 1.** Power to estimate 50% total vaccine efficacy for SOR and SORI parameter estimates for contact times of 1, 2 and 3 years Statistical power as a function of group size to estimate a significant difference in vaccinated and unvaccinated groups at the 95% level for an effect size of 50% total vaccine efficacy based upon SOR and SORI parameter estimates.

DOI: https://doi.org/10.7554/eLife.27694.035

**Appendix 2—Figure 2 supplement 2.** Power to estimate 25% total vaccine efficacy for SOR and SORI parameter estimates for contact times of one, two and three years Statistical power as a function of group size to estimate a significant difference in vaccinated and unvaccinated groups at the 95% level for an effect size of 25% total vaccine efficacy based upon SOR and SORI parameter estimates.

DOI: https://doi.org/10.7554/eLife.27694.036

Two experimental groups with 4N co-housed animals; 2N seeder reactor animals; N vaccinated sentinel animals and N unvaccinated sentinels; 60 day interval DIVA testing; Retention of test-positive animals; Two phase design with sentinel animals from Phase I used as seeder animals in Phase II

The basic experimental unit of this design (**Appendix 2—figure 1**) is a group where an equal number of (presumed infected) seeder animals ($S$) are co-housed with susceptible sentinel animals ($U, V$) that also differ with respect to their vaccination status (Vaccinated $V$, Unvaccinated $U$). The first experimental phase (Phase I) consists of two experimental replicates (Group 1 and Group 2) that essentially compare the relative rate of transmission of *M. bovis* to unvaccinated and vaccinated sentinel animals when challenged by naturally infected seeder animals ($S_R$) recruited from the field. Any reduction in the rate of transmission to the vaccinated group in Phase I will have contributions from the direct protection offered through a reduction in susceptibility and the indirect effect of the reduction in susceptibility and infectiousness of the sentinel animals.

Taken alone, and in common with past experimental estimates of the efficacy of BCG, the two experimental groups in Phase I can only estimate the direct efficacy of vaccination. Phase II provides information on the effect of vaccination on infectiousness by using the exposed sentinel animals from Phase I as seeder animals for a second round of transmission with fresh vaccinated ($V$) and unvaccinated ($U$) sentinels. This comparison is essential to separate the relative effect of vaccination on susceptibility and infectiousness. Note that we can distinguish the (indirect) effect of reduced infectiousness from the indirect effect of reduced susceptibility

because we use the information regarding which animals are infected. In contrast to field trials, where the extrinsic force of infection acting on different herds is completely unobserved, the number of seeder and test-positive animals can be used as a first estimate to adjust for the change in the force of infection over the course of an experiment. Thus, the rate of transmission of bTB to sentinel animals in both vaccinated and unvaccinated groups can be directly estimated via the rate at which animals become DIVA positive, rather than relying on relative risk ratios based on the clinical end-point. This is a key advantage which opens up the use of more powerful methods of analysis such as survival or mechanistic chain-binomial models (Velthuis et al. 2007).

## Sample size calculations for natural transmission study to evaluate vaccine efficacy

For natural transmission studies for a chronic infection such as bTB, we can use the reproductive ratio R as a design parameter. R can be manipulated directly (at least within the bounds set by the bovine lifespan) through setting the in-contact time ($T_C$) of each experimental phase. The basic requirement for the design of a natural transmission study is that we see transmission in all experimental groups during both experimental phases. Thus, we must ensure that the reproductive ratio R is greater than one and ideally is large enough that sufficient seeder animals are generated in Phase I to be used for Phase II.

The sample size calculations to power the proposed transmission study are conceptually equivalent to the comparison of two treatments for which the power can be readily estimated. The statistical comparison of two treatments with respect to transmission can be based on the final size (FS) for the two transmission chains in the different treatment groups, or on the measurements of the number of infected and susceptible at the beginning of an interval and the number of cases in each interval (*Velthuis et al., 2007a*, *2007b*). Clearly the latter methods have a higher power as the time-series of transmission events provide more information than the final size alone. However, for the calculation of the sample size we want to use a method that gives a conservative estimate of the sample size (too large rather than too small). Thus, the FS methods are more suitable for this purpose.

The power of experiments analysed by FS is approximately the same irrespective on how animals are distributed over groups, provided that, as in our designs, we aim for a 50:50 mix of seeder (infected) and sentinel animals in each transmission group (*Velthuis et al., 2007a*). Thus 20 groups with pairs (one seeder and one sentinel) has a similar power as 1 experiment with 40 animals (20 seeder and 20 sentinels). The statistical analysis with FS depends on finding the joint final size probability distribution for all the groups. For larger groups this is difficult but for pairs this is straightforward. Each pair has only two possible outcomes: the recipient becomes infected or not. This depends on what happens with the infected animal: it either recovers first or it infects the recipient first. The events that can happen are:

$$(S,I) \rightarrow (S-1, I+1) \text{ with rate } \beta \frac{S\,I}{N}$$

$$(S,I) \rightarrow (S, I-1) \text{ with rate } \alpha\,I$$

The probability that the infection occurs first (between now and infinity, or in other words given that one of the events occurs) is (competing risks):

$$\frac{\beta \frac{S\,I}{N}}{\beta \frac{S\,I}{N} + \alpha\,I}$$

Which simplifies to assuming $I \neq 0$ and the basic reproduction ratio $R = \frac{\beta}{\alpha}$:

$$\frac{R\,S}{R\,S + N}$$

Thus, for a pairwise experiment S = 1 (I = 1 check not 0) and N = 2:

$$\frac{R}{R+2}$$

The other possibility: that recovery occurs before the contact infection is of course the complementary probability. Thus each pairwise experiment is one Bernoulli experiment with $p = \frac{R}{R+2}$.

Thus n multiple pairwise experiments follow a binomial distribution with this p and n as total number of trials.

Sample size can be calculated based upon this information, either numerically using Fishers exact test or analytically using an asymptotic normal approximation. When $n$ is large ($np>0.5$, $n(1-p)>5$) the binomial distributions can be approximated by normal distributions leading to the following expression for n:

$$n = (p_1(1-p_1) + p_2(1-p_2)) \left( \frac{Z_\alpha + Z_\beta}{p_1 - p_2} \right)^2$$

which is in our case the number of pairs thus:

$$n = 2\,(p_1(1-p_1) + p_2(1-p_2)) \left( \frac{Z_\alpha + Z_\beta}{p_1 - p_2} \right)^2$$

and where $Z_\alpha$ and $Z_\beta$ are the critical values of the standard Normal distribution for the two types of error, e.g. $Z_\alpha = 1.96$ (one sided, error rate 2.5%) and $Z_\beta = 0.842$ (power 80%), and the two values are based on the reproduction ratios under the alternative hypothesis (i.e. when there is a difference), $p_1 = \frac{R_1}{R_1+2}$ and $p_2 = \frac{R_2}{R_2+2}$.

In order to proceed we require an estimate of the reproductive ratio ($R$) for bTB in an experimental transmission setting. We define this experimental reproductive ratio $R$ as the expected number of infections when a single infectious seeder animal is placed in contact with a completely susceptible population of size $H$ for an in-contact time $T_C$. The definition is in contrast to the basic reproduction ratio ($R_0$), which for a chronic infection such as bTB would be the expected number of infections over the lifetime of a single infectious seeder animal.

## Calculated sample size and experimental duration

For a balanced design with an equal number of seeder and incontact animals the sample size therefore depends on two parameters – the number of pairwise comparisons between sentinel animals (or equivalently the group size) and the average expected numbers of infections per seeder animal over the course of the experiment (the experimental reproduction ratio $R$).

For a chronic infection like bovine Tuberculosis, $R$ is essentially a design parameter which can be adjusted by adjusting the duration of the in-contact period between seeder and sentinel animals. The statistical power to estimate a significant difference between vaccinated and unvaccinated animals is relatively insensitive to the value of $R$ provided it is greater than the threshold value of 1. For an $R \sim 1.5 - 2.0$ a group size of 52 animals would provide 80% power to estimate an effect size (total vaccine efficacy) of 75% at the 95% significance level (*Appendix 2—table 1*). Reducing the effect size to 50% and 25% would increase the required group size to 128 and 600 animals respectively (*Appendix 2—table 1*).

**Appendix 2—table 1.** Group sizes to achieve at least ~ 80% power for constant experimental $R$.

| Effect size (Total Vaccine Efficacy) | Experimental reproductive ratio R | Group size | Power |
|---|---|---|---|
| 75% | 2.0 | 52 | 82% |
| 75% | 1.5 | 52 | 80% |
| 50% | 2.0 | 128 | 82% |
| 50% | 1.5 | 128 | 80% |

*Appendix 2—table 1 continued on next page*

*Appendix 2—table 1 continued*

| Effect size (Total Vaccine Efficacy) | Experimental reproductive ratio R | Group size | Power |
|---|---|---|---|
| 25% | 2.0 | 600 | 82% |
| 25% | 1.5 | 600 | 80% |

DOI: https://doi.org/10.7554/eLife.27694.037

Predicting the in-contact period ($T_C$) necessary to achieve this target value of $R \sim 1.5$ is challenging given the sparsity of quantitative estimates of transmission rates of bTB in a small group setting. A natural transmission experiment was attempted at the Animal and Plant Health Agency (APHA) facilities at Weybridge using GB reactor animals. However, specific limitations of the animal housing and lack of time-series data make interpretation of this data challenging – and posterior distributions for the estimated rate of transmission range over several orders of magnitude (see Appendix 3).

Robust estimates of transmission rates are almost exclusively derived from population level field data, the relevance of which for an experimental controlled transmission setting is questionable. There is considerable variability between estimates of the rate of cattle-to-cattle transmission from the literature (a brief review is presented as Appendix 3). A further challenge for application to small group settings is that field estimates of transmission demonstrate a clear increase in the rate of transmission of bTB with herd size.

In *Appendix 2—figure 2* we explore the impact of density dependence on the likely duration of natural transmission studies based upon the predicted experimental reproduction ratio $R$ from our two most recent within-herd transmission models (*Conlan et al., 2015*). For an effect size equivalent to a 75% total efficacy of vaccination and a 1 year contact time, a group size of between 80 (SORI model) and 150 (SOR model) animals would be necessary to achieve an 80% power at the 95% significance level. For a two-year contact time these samples sizes fall to between 50 and 100 animals for the SORI and SOR models respectively. For a 1 year duration sample sizes increase to between 200-350 animals for a 50% total efficacy (*Appendix 2—figure 2–figure supplement 1*) and in excess of 800 animals for a 25% total efficacy of vaccination (*Appendix 2—figure 2–figure supplement 2*).

## Appendix 3

DOI: https://doi.org/10.7554/eLife.27694.038

# Mini-review of estimated cattle-to-cattle transmission rates of bTB

The duration of a challenge study depends critically on our estimate of the likely rate of cattle-to-cattle transmission. A pilot natural transmission study with vaccinated and unvaccinated animals was carried out by the Animal Plant Health Agency (APHA, formally the Animal Health and Veterinary Laboratory Agency, AHVLA) at Weybridge in the UK. Unfortunately, for reasons discussed below, this study does not provide useful estimates of transmission rates. There is therefore a need to place these findings in the wider context of bTB transmission estimates from the literature to inform the likely range of transmission rates we might expect to be able to achieve in new experimental designs.

## Weybridge pilot study and other natural transmission experiments

A pilot natural transmission study was carried out by APHA in Weybridge using 40 reactor animals recruited from UK herds and 60 sentinel animals to give a study population of 100. Animals had to be grouped together in pens of 10 animals (6 sentinels and 4 reactor animals) due to the physical design of the barn (*Khatri et al., 2012*) which is likely to have limited the potential for transmission. Only 8 transmission events were observed after an in-contact period $T_C$ of 12 months. With no time series information, this final size distribution is the only information from which transmission rates can be inferred.

We estimated the (frequency-dependent) transmission parameter ($\beta$) for this final size data using an exact likelihood for the stochastic SI model calculated by numerically solving the master equations (*Allen 2003*). The stochastic SI model has a single event with rate $\beta S \frac{I}{N}$, where $S$ is the number of susceptible sentinel animals, $I$ the number of infected animals, $N = S + I$ the size of the group.

The point (maximum likelihood) estimate of $2.24 \times 10^{-5}$ per year from this data is exceptionally small compared with an estimated frequency dependent transmission parameter of the order of 2 per year from field data (*Fischer et al., 2005*; *Barlow et al., 1997*). However, given the limited information in the final size distribution the 95% credible interval of the posterior estimate (*Appendix 3-Figure 1*) ranges over several orders of magnitude (95% CI: $1.4 \times 10{-14} - 5.97$). The consistency of this estimated effective (i.e. frequency dependent) rate of transmission with density dependent estimates of bTB transmission from the field depends on what we interpret as the herd size for this population (*Appendix 3-Figure 1*). The maximum likelihood estimate from the Weybridge study is more consistent with an effective group size of 10 (the size of holding pens) that a group size equivalent to the size of the barn (100 animals). While this is suggestive that rates of transmission may be enhanced in a more suitable facility which allows for free mixing of animals, we must be cautious given that the full posterior predictive distributions from field estimates like within the posterior estimate of transmission from the Weybridge study.

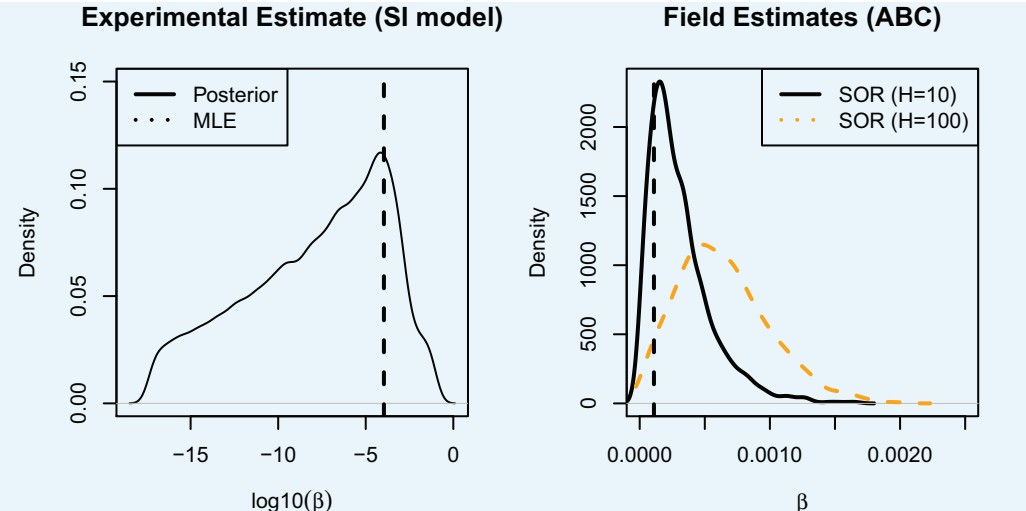

**Appendix 3—figure 1.** Posterior estimates of rates of cattle-to-cattle transmission from Weybridge pilot study (Left) Posterior distribution and maximum likelihood estimate for frequency dependent transmission rate ($\beta$) calculated from final size distribution (*Khatri et al., 2012* (Right) Posterior predictive distributions for effective (frequency dependent) transmission parameter ($\beta$) from field estimates of the SOR transmission model with an assumed herd size of 10 (black) and 100 (orange, dashed). MLE estimate from Weybridge pilot study is show by vertical black dashed line.

DOI: https://doi.org/10.7554/eLife.27694.039

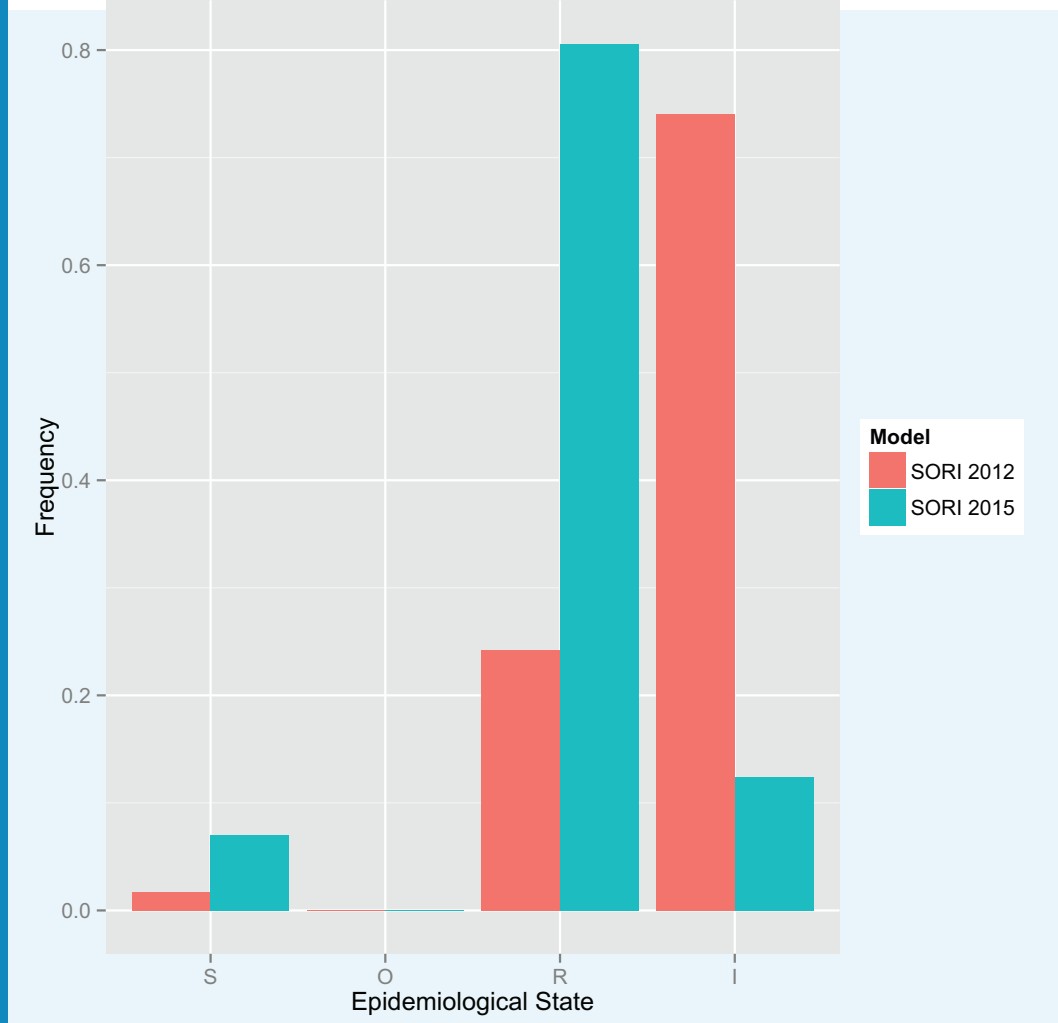

**Appendix 3—figure 2.** Simulated probability of a randomly selected reactor being in the S, R or I compartment for the SORI 2012 and SORI 2015 models Predictive distributions for the probability of reactor animals being in the S, O, R and I compartments for the SORI 2012 and SORI 2015 models. For consistency, these proportions are used as initial conditions for 'reactor' animals used as seeders in Phase I for the respective model scenarios. The true proportion of individuals in each epidemiological state will depend on the distribution of latent periods – which is unknown – but assumed to be exponential for these models (constant rate of progression).

DOI: https://doi.org/10.7554/eLife.27694.040

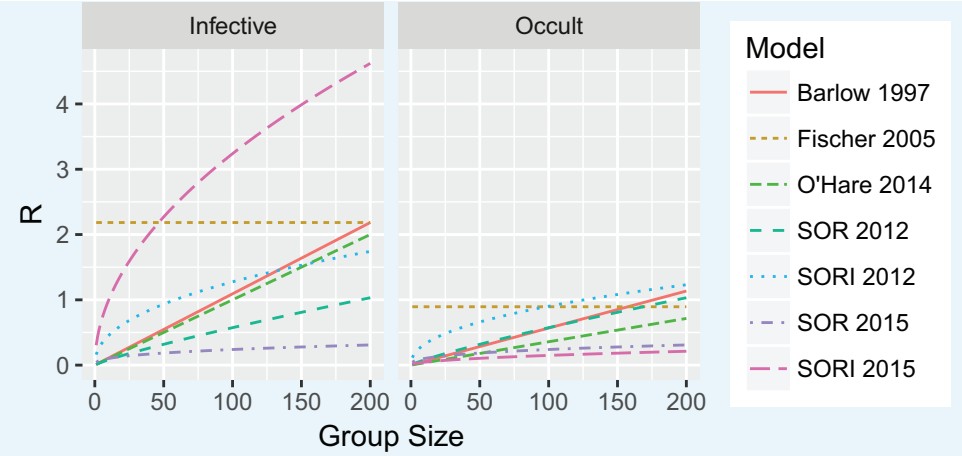

**Appendix 3—figure 3.** Point predictions for the predicted experimental $R$ for a 1 year transmission experiment from field estimates of bTB transmission rates (**Appendix 3—table 1**).

DOI: https://doi.org/10.7554/eLife.27694.041

Despite the lack of success of this experiment, there are further good reasons to expect that higher rates of transmission are achievable in designs that balance the number of infectious and susceptible contacts within each group (which as noted before was limited in the pilot experiment due to facility constraints). Indeed, experimental studies in endemically infected countries, such as Ethiopia, have achieved higher rates of transmission (**Ameni et al., 2010**). This success has since been repeated with nearly identical patterns of transmission to the published study (Ameni and Vordermeier, unpublished data). Unfortunately, rates of transmission have not yet been quantified empirically from these studies.

The imprecision of the transmission parameter estimate from (**Khatri et al., 2012**) makes it unsuitable to define a prior range of estimates to design any future experimental transmission studies. We therefore look to field estimates of cattle-to-cattle transmission to provide a more appropriate basis to progress. This route requires careful discussion as estimates vary considerably between different studies and populations. Furthermore, transmission rates in the field and in an experimental setting cannot necessarily be expected to be equivalent due to differences in the management, health and welfare of animals and other extrinsic factors acting on herds. However, in the absence of other alternatives, field estimates of bTB transmission provide the only reasonable basis to proceed.

Published models of bovine Tuberculosis in cattle share a common basic structure but differ in two key aspects with respect to the assumed progression of infected animals to infectiousness (latency) and the scaling of transmission rates with herd size (so-called density dependence).

## Latency of bovine Tuberculosis

Tuberculosis is a chronic and progressive infectious disease that has been described as having an incubation period that ranges from a few weeks to a lifetime (Comstock, Livesay, and Woolpert 1974). Within-herd transmission models have been primarily concerned with assessing the efficacy of test-and-slaughter protocols. Taking a standard compartmental approach, these models subdivide the period of epidemiological latency between infection and infectiousness into two further compartments based on the animals reactivity to the tuberculin skin test. When susceptible ($S$) animals become infected within such models they enter an "Occult" ($O$) compartment where they are insensitive to testing, before progressing to become "Reactive" ($R$) to the skin test and eventually (reactive and) "Infectious" (I). Such SORI models typically assume the average time between infection and infectiousness is of the order of 1 year or longer.

However, due to the basic uncertainty between diagnostic status and infectiousness for bTB discussed earlier, this period of latency has not been directly estimated and is poorly identified in models estimated from population level data (*Conlan et al., 2012*; *O'Hare et al., 2014*; *Brooks-Pollock et al., 2014*; *Conlan et al., 2015*). Evidence from experimental challenge studies (*Kao et al., 2007*) and the rates of detection of visible lesions in young reactor animals (*Brooks-Pollock et al., 2013*) suggest that this duration of latency has the potential to be short in some contexts. For this reason, but primarily for the sake of parsimony, Conlan et al. estimated simpler alternative SOR models where animals are potentially infectious immediately upon infection and the latent stages only affect the reaction of animals to the diagnostic skin test (*Conlan et al., 2012*, *2015*). Model fits to field data have not thus far provided any basis to choose between these two structures that make very different predictions for the rates of cattle-to-cattle transmission within herds, the burden of infection remaining after herds clear movement restrictions and the scaling of transmission rates with herd size (*Conlan et al., 2012*).

## Scaling of rates of cattle-to-cattle transmission with herd size

An empirical relationship between herd size and the abundance of bovine TB has been described in a variety of contexts since well before the introduction of systematic control measures for bTB (Francis 1947). This empirical relationship has motivated bTB models to model transmission using a so-called 'density' dependent transmission function (*Barlow et al. 1997*; *Biek et al. 2012*; *O'Hare et al. 2014*). This terminology is somewhat of a misnomer as under this assumption transmission (and thus the basic reproduction ratio $(R_0)$ is actually assumed to scale with herd size rather than the true density of animals (*De Jong, Diekmann, and Heesterbeek 1995*; *Begon et al. 2002*). The biological mechanisms responsible for generating such a herd size dependence are unclear, motivating other authors (Fischer et al. 2005) to use the more theoretically appealing and justifiable assumption of frequency dependent transmission (where transmission rates scale independently of herd size).

Conlan et al. introduced a non-linearly density dependence transmission function (*Melegaro et al., 2004*; *Smith et al., 2009*) to attempt to select between these two extreme assumptions (*Conlan et al., 2012*) where the rate at which susceptible individuals within a herd becomes infectious is modelled as:

$$\frac{\beta I}{\left(\frac{H}{H_m}\right)^q}$$

$I$ is the number of infectious individuals (taken as the sum of the $O$ and $R$ compartments for a SOR model) in a herd of size $H$. $H_m = 165$. $q$ is a constant defining a centering transformation used to improve convergence of parameter estimation. Finally, $q$ measures the strength of dependence of transmission rates of $H$, with $q = 0$, corresponding to density dependence and $q = 1$ frequency dependence.

## Comparison of field estimates of cattle-to-cattle transmission rates

*Appendix 3—table 1* summarises the point estimates of transmission rates, occult and reactive periods from seven relevant models from the literature. The estimates by Barlow and Fischer were calculated (rather than formally estimated) based on the same observation of transmission within a single herd (of 200 cattle) in New Zealand but make opposite assumptions about density dependence (*Barlow et al., 1997*; *Fischer et al., 2005*). Conlan and O'Hare (*Conlan et al., 2012*; *O'Hare et al., 2014*; *Conlan et al., 2015*) used national GB data to estimate their models using approximate Bayesian methods of inference. The two variants of the SOR and SORI models published by Conlan et al. in 2012 and 2015, differ primarily with respect to the demographic structure of herds and the prior distributions used for estimation of the occult and reactive periods. The 2012 variants used a particularly simple demographic model for herds with an exponential age distribution (*Conlan et al., 2012*). The 2015 variants implement the SOR and SORI models as individual based models with a realistic

age-structure reconstructed from cattle tracing system (CTS) records (**Conlan et al., 2015**) and incorporate an age-dependent risk of infection and detection of visible lesions (**Brooks-Pollock et al., 2013**). The 2015 variants are also distinguished, in common with (**Brooks-Pollock et al., 2014**), by using a far less restrictive prior distribution on the duration of the reactive period leading to far greater estimates for this parameter than previous models.

**Appendix 3—table 1.** Point estimates of rates of cattle-to-cattle transmission of bTB.

| Model | Transmission parameter $\beta$ | Transmission parameter $q$ | Occult period $T_O$ | Reactive period $T_R$ |
|---|---|---|---|---|
| Barlow et al., 1997 | $3 \times 10^{-5}$ days$^{-1}$ | 0 | 22 days | 180 days |
| Fischer et al., 2005 | $6 \times 10^{-3}$ days$^{-1}$ | 1 (Note $H_M = 1$) | 41 days | 233 days |
| O'Hare et al., 2014 | $2.7 \times 10^{-5}$ days$^{-1}$ | 0 | 100 days | 190 days |
| Conlan et al. (2012) | $1.5 \times 10^{-5}$ days$^{-1}$ | 0.15 | 4 days | NA |
| Conlan et al. (2012) | $2.7 \times 10^{-5}$ days$^{-1}$ | 0.56 | 28 days | 77 days |
| Conlan et al. (2015) | $4.9 \times 10^{-5}$ days$^{-1}$ | 0.63 | 63 days | NA |
| Conlan et al. (2015) | $7.2 \times 10^{-5}$ days$^{-1}$ | 0.49 | 73 days | 2,572 days |

DOI: https://doi.org/10.7554/eLife.27694.042

Although the majority of these studies estimated full posterior distributions for model parameters, for the purpose of comparison only point (median) estimates are discussed here. The transmission parameter in-itself has no straightforward biological interpretation, as the potential for transmission also depends on the estimated values of $q$, $T_O$ and $T_R$. It is therefore more appropriate to compare estimates of transmission through the corresponding reproductive ratio for a defined population.

As previously discussed, the reproductive ratio for an experimental group ($R$) will depend on the group size (H) and in-contact time between seeder and sentinel animals ($T_C$). $R$ can be calculated as a function of the transmission parameters through the next generation operator (De Jong, Diekmann, and Heesterbeek 1995). For the SOR model $R$ is independent of the occult and reactive periods:

$$R = \frac{\beta T_C}{\left(\frac{H}{H_m}\right)^q}$$

while for the SORI model, R also depends on the probability that a seeder animal is in the occult ($p_O$), reactive ($p_R$) or infectious ($p_I$) compartment:

$$R = \frac{\beta}{\left(\frac{H}{H_m}\right)^q} \left( p_I T_C + \frac{p_R}{\sigma_R}\left(e^{-\sigma_R T_C} - 1\right) + \frac{p_O}{\sigma_O - \sigma_R}\left(\frac{\sigma_O}{\sigma_R}\left(e^{-\sigma_R T_C} - 1\right) - \frac{\sigma_R}{\sigma_O}\left(e^{-\sigma_O T_C} - 1\right)\right) \right)$$

$\sigma_R = \frac{1}{T_R}$ where $\sigma_O = \frac{1}{T_O}$.

In **Appendix 3—figure 1** we compare the predicted (point estimates) of $R$ for a contact time of 1 year using the point estimates from **Appendix 3—table 1**. For SORI type models, the predicted $R$ depends heavily on whether we assume the initial infected individual is either latently infected (Occult, right panel) or infectious (Infective, left panel). For the purpose of powering these designs we make the conservative assumption that we will only be able to recruit seeder animals based upon SICCT test status giving the empirical distributions summarised in **Appendix 3—figure 2**. In practice, we would hope to be able to increase the proportion of seeder animals in the infectious class by the requirement that they are both

SICCT and IGRA positive, or through the use of more sophisticated assays (IL2 or micro-RNA expression) that predict more advanced disease.

*Appendix 3—figure 3* clearly illustrates the importance of how transmission rates scale with herd size with respect to the design of transmission studies for bTB. With frequency dependent scaling of transmission rates a 1 year contact time will be comfortably sufficient to ensure that R is greater than the epidemic threshold (1) for the smallest group. However, for density dependent estimates this will only be true for group sizes in excess (for some parameter sets far in excess) of 50 animals. There is considerable variability in the assumed (or estimated) scaling of $R$ with group size between different models, although some commonalities emerge. In particular we note that the predicted $R$ for the traditional SORI model structures (Barlow et al. 1997; *Fischer et al., 2005*; O'Hare et al. 2014) are comparable in large herds of ~ 200 animals. The predicted $R$ for models without epidemiological latency (SOR 2012, SOR 2015) is consistently less the more traditional SORI models. Finally, the SOR 2015 and SORI 2015 models demonstrate the greatest discrepancy between each other and the rest of the estimates, but also differ considerably with respect to both model demography and the prior assumptions used for their estimation (Conlan et al. 2015). The variability in these predicted estimates of $R$ demonstrate the challenges of translating parameter estimates from the field – which are in turn also dependent on estimates of the efficacy of SICCT testing – to an experimental setting.

Given this variability selecting a single estimate to power and benchmark designs would be inappropriate. For convenience, as these models are directly available to us, we define a set of scenarios based upon estimates from the most recent SOR 2015 and SORI 2015 models (*Conlan et al., 2012*, *2015*). The scaling of transmission rates with herd-size is an empirical observation and may well arise due to husbandry or herd management factors that are directly correlated with herd-size that may well not apply to an experimental setting. If this is the case then density dependent estimates from the field may underestimate the potential for transmission in small groups. To allow for this possibility we define a set of additional "Optimistic" scenarios where transmission rates are fixed to be independent of herd size. We achieve this by fixing the effective herd size to $H_m = 165$, where the SORI model estimates approach the upper frequency dependent estimate of $R \sim 2.0$ (*Fischer et al., 2005*).

