## [Decision Letter]

Thank you for submitting your article "The intractable challenge of evaluating cattle vaccination as a control for bovine Tuberculosis" for consideration by *eLife*. Your article has been favorably evaluated by Tadatsugu Taniguchi (Senior Editor) and three reviewers, one of whom is a member of our Board of Reviewing Editors. The reviewers have opted to remain anonymous.

The reviewers have discussed the reviews with one another and the Reviewing Editor has drafted this decision to help you prepare a revised submission.

All three reviewers thought the work to be important and of broad interest, but each raised a number distinct and important major issues which need to be addressed in a revised submission. In addition, all reviewers found the current manuscript overlong and lacking in clarity. We would urge greater selectivity in presenting key results, and a substantial overhaul (and shortening) of the main text of the paper. Additional analyses and detailed methods can be included as supplementary information.

The reviewers also raised concerns about how study power was considered, and the issues of likely attack rates in transmission experiments and extrinsic infection rates should also be given particular consideration, in addition to the other detailed comments made.

*Reviewer #1:*

This paper explores an important topic – how to test the efficacy of a bovine TB vaccine – using simulation models of TB transmission to test different trial designs. Overall, the analysis seem rigorous to the extent I could judge it, but ambiguity or missing details left me with several significant questions. The presentation of the paper is far from ideal overall – it is much too long, difficult to follow, and lack of required detail in the Materials and methods means it is not always clear how results have been derived. Really the paper needs major reorganisation, with a (much) shorter and selective main text, then a more detailed supplement to detail methods, parameterisation and sensitivity analyses.

Issues:

- Subsection “Review of estimated cattle-to-cattle transmission rates for bovine TB” is critical – the failure of the AHVLA experiment fundamentally calls into question the feasibility of experimental studies of vaccine efficacy in the UK context. This really needs to be highlighted up front in the paper (not just in the Materials and methods). I note the authors contention that increasing group size and the proportion of the group initially infected would likely substantially increase infection rates, but given the cost of such studies and the need to demonstrate that the results will likely transfer to the field setting, I would suggest that funding any large-scale transmission study should be contingent upon achieving a higher attack rate in a small pilot (e.g. 50 animals, 25 infected, 25 uninfected). Such a pilot would also give invaluable data with which to better power future vaccine studies.

- How exactly are the expected values plotted in Figure 3A (and like figures) being estimated? No details are given in the Materials and methods. I'm guessing it is the average of the estimated VE over a large number of simulations of the experimental transmission study of X herds? How was VE calculated – using the expressions given in Figure 1? Giving some representation of the 95% range of the estimates from single experiments would be useful.

- Likewise, how exactly are the estimates of power calculated in Figure 3B? I presume from their simulations? In doing so, are they also simulating a 2-level analysis of the simulated trial results (i.e. accounting for variation between herds)? What primary end-point is being evaluated – a difference in attack rates between vaccinated and control herds, or estimation of VE to some level of precision (e.g. +/-0.05). I would suggest the latter is more useful. i.e. for a fixed set of assumptions about VE, run 1000 simulations of the trial in X herds, and count the number of simulations, Z, for which the desired measure of efficacy is estimated to within +/-0.05 (say) of its true value. Power is then Z/1000.

- "Clearly the latter methods have a higher power and that is the method that is used for estimation and calculation of the post-hoc power from the simulation studies described below (A4)". I am confused. What does A4 refer to? Which tables and figures use the FS based approximate analytical calculation and which use direct simulation? Precise details of how vaccine efficacy and power was calculated from simulations of the experimental design should be given. I don't frankly see that the analytical approach in the subsection “Sample size calculations for natural transmission study” adds anything very much. Given any experiment must have a fixed duration, what is important is the net attack rate seen in the vaccinated and control animals over the duration of the experiment. This depends on the transmission rate (and how that varies as a function of the time from infection) and the duration of the study phases (i.e. contact time). Going from transmission rates to R and back confuses things, at least for me. I would rather see Figure 10 show the posterior distribution for transmission rate (including the herd size factor), therefore. This would be more informative than the estimates given in Table 12. It would also allow Figure 9 to be removed – which doesn't add anything informative beyond Figure 11 in my view – indeed the addition of the red vertical dashed lines to Figure 9 is confusing.

- Continuing in the same vein, the paper gives the impression that Table 11 is driven by the results of Table 10, which misses the subtleties of Figure 11. This latter figure is the most interesting in the paper, but I didn't understand some of the trends in Figure 11. Why does increasing contact time decrease power (for fixed group size) for some model variants and vaccine efficacies? I can only assume this is because infection rates are saturating in both groups. However, if the experiment was analysed making use of all the DIVA test results in a survival analysis, this shouldn't matter – the higher infection hazard in the control group animals should still be resolvable in the first phase, and the lower infectiousness of the vaccinated animals in the second phase. As I've said above, the authors need to give precise details of how power is being estimated from the simulation for Figure 11 (see above) – how are the (simulated) experimental results being analysed, what is the trial primary end-point (i.e. what statistical test is being examined when calculating power)? Again, given the cost of such experiments, analysis needs to make best possible use of the data collected – which survival analysis is more likely to achieve than simple comparison of final attack rates.

- In the first paragraph of the subsection “Group size and duration of transmission studies under different transmission scenarios” – What are Figures B3 and B4 and what is Table B1? Assuming Table B1 is actually Table 12, how are the values given in that table used to generate Figure 9 on? In particular, were the first 2 rows of Table 12 used for any simulations, or just the Conlan et al. estimates? As mentioned elsewhere, I would drop the Conlan 2012 results – presumably they were superseded by the 2015 ones, and they give rather optimistic results for the trial contact times.

- Table 11, 25% effect size – the bottom-most rows have a group size of 800, while this group size isn't mentioned in Table 10. Is this a typo? As commented below, I don't feel including all the 2012 model variants adds anything here.

- The second paragraph of the subsection “Comparison of field estimates of cattle-to-cattle transmission rates” is unnecessary. Presumably the 2015 models are preferred, so reference to and results relating to the 2012 model (half of Figure 8, Figures 9, 11, Figure 11—figure supplements 2, 4, half of Table 11) can be removed.

*Reviewer #2:*

This paper is a well written, thorough and painstaking analysis of a narrow technical issue, sample size calculations for a hypothetical trial of vaccination to protect cattle from bovine TB. It is a substantial and technically useful piece of work, though not easily generalizable given the specific and complicated details of bovine TB epidemiology and management in the UK.

I agree with the statement (subsection “Conceptual design to estimate vaccine efficacy and herd level effectiveness”, fifth paragraph) that it is important to evaluate the impact of vaccination on infectiousness as well as susceptibility – as a rule the former is ignored.

The prediction that there would be, in the situation modelled, only a very small indirect impact of vaccination means that very large sample sizes would be needed for a trial to detect it. The situation modelled includes current test and slaughter practices, but presumably if a vaccine were to be used it would not be used in conjunction with these. If it was, as is spelt out later, the additional benefit would be very small and presumably not cost effective. Or is that the point the authors wish to make? Either way, a vaccination-only scenario would be of interest (regardless of current EC requirements).

The way statistical power is estimated (more than "slightly" unusual in my view – Discussion, third paragraph) also makes the study less generalizable. It would help to set out what efficacy we are looking for, greater than zero seems a very low bar. (What's more, the subsequent discussion about the DIVA test implies that even if the vaccine itself had zero efficacy there would be some effect of the more sensitive test – Discussion, sixth paragraph). These impacts could be partitioned by varying parameter settings appropriately.

The role of extrinsic infection (Discussion, eighth paragraph) could also be explored more systematically. The issue of testing efficacy locally when the ultimate aim would be to intervene over a whole population is problematic for many intervention trials for infectious disease.

Overall, I felt that, though a rather daunting volume of results are presented already, more could have been done to dissect out the likely multiple contributors to the low efficacy anticipated in a herd level vaccination trial.

*Reviewer #3:*

General Comments/Suggestions:

This manuscript describes the adaptation of previously published models to understand how field trials might (or might not!) detect the benefits of a bTB vaccine deployed in Britain. In general, the methods and conclusions seem sound, and represent an important warning on relying on these sorts of trials. I did have some problems understanding the interpretation results shown in the figures associated with this manuscript, and I suspect that some figures may not be referred to correctly. I have included some suggestions below on how to make the figures more easily readable. The mathematical typesetting in the manuscript also made it somewhat less readable – typesetting that sets apart mathematical entities like R and R_0_ more clearly would have helped me, and would also have limited confusion when "R" is used both as a reproduction parameter, and as a compartment (e.g. in the SORI model).

Because this work is based on mathematical modelling, I would urge the authors to make as much of the modelling code as possible available on a public repository, or provide links to the previously-published code used. Publishing code in this way makes the work much more reproducible.

[Editors’ note: the authors were asked to provide a plan for revisions before the editors issued a final decision. What follows is the editors’ letter requesting such plan.]

Thank you for sending your article entitled "The intractable challenge of evaluating cattle vaccination as a control for bovine Tuberculosis" for peer review at *eLife*. Your revised article has been favorably evaluated by Tadatsugu Taniguchi (Senior Editor) and three reviewers, one of whom is a member of our Board of Reviewing Editors.

Please review the major comments of reviewer #1 (the Reviewing Editor), which center on the interpretation of your results and their relevance for policy. We would then ask you to respond within the next two weeks with your views on how justified you feel these comments to be, and an action plan and timetable for the completion of any additional work. We plan to share your responses with the reviewers and then issue a binding recommendation.

*Reviewer #1:*

The rewritten paper is much clearer and more comprehensible. My few technical comments are detailed below. I do have more major issues with the conclusions and tone of the paper however:

- Given the problematic experience with previous transmission experiments, my own conclusion from reading this paper was that relatively small field trial of 200 herds would give valuable information about the likely veterinary health impact of vaccination at the individual animal level (but that a trial with 500 herds would be needed to measure herd-level effectiveness).

- Data from an experimental study is fundamentally different in quality than that from a randomised field trial. In human public health, the former might be viewed as equivalent to a pre-phase II human challenge study, which gives proof of concept. It does not guarantee that the results can be read across to the natural setting – which is why phase III trials are still needed.

- The authors seem unnecessarily pessimistic about what their simulations imply for the feasibility of field trials, at least at the individual animal level. I interpret Figure 3 as showing that a trial run in 200 herds would have excellent power at measuring the direct effect of vaccination, and reasonable power at measuring the 'total' effect.

- Yes, measuring indirect effects is difficult, but arguably is addressable in a post-marketing (phase IV) implementation study.

- I think there are issues with a cluster-randomised trial with 50% coverage in each cluster (herd). Even for the vaccinated animals, outcomes will be different in a herd with 100% vs 50% coverage of a leaky vaccine. Plus, presumably the goal for any widescale vaccination policy would be 100% coverage? A 3-arm trial with herd level vaccination coverage of 0, 50% and 100% in the three arms might be more informative. Comparing attack rates in the 0 and 100% arms would give a measure of total effect (the most important outcome). Comparing vaccinated animals in the 50% and 100% coverage arms and unvaccinated animals in the 50% and 0% arms would give more information (and therefore power) to differentiated impacts of vaccination on infectiousness and susceptibility.

- Appendix 2 on whole-herd effectiveness is interesting and critically important (to the extent I would much rather see this in the main text and the discussion of experimental transmission studies in an appendix) – and in my view calls into question the whole viability of vaccination, if the overall impact on herd breakdowns is really only likely to be in the range of 10-20%. Putting that (major) issue to one side, the results in this section also highlight the potential benefits of a 3-arm design.

- Regarding the discussion of bias in RR measures – it is unsurprising that such measures underestimate ε_s_ – it's the difference between comparing a hazard with a cumulative hazard.

- Indeed, Figure 3 (and the supplementary version) seems to show that one could use models to quite reliably go back from the measured relative risks to the underlying effect of vaccine on susceptibility – albeit not on infectiousness. – Figure

*Reviewer #2:*

The first round of reviews seems to have picked up a large number of errors and presentational issues. The authors have addressed these fairly comprehensively and the manuscript is greatly improved as a result. If the topic is thought appropriate for *eLife* then I recommend that the manuscript is now acceptable for publication.

*Reviewer #3:*

In general I am satisfied with the re-organisation and changes made. I find the manuscript is more focused and easier to get through in its current form. I still find the Appendices somewhat arduous, and would encourage the authors to consider any last-minute changes they can make to streamline them, but I accept that sometimes Appendices with technical content can be long.

I was a bit disappointed that the authors felt they could not address the impact of the distribution of latencies on the design, but accept their justification that, with the very high level of uncertainty on these distributions, they do not want to "muddy the waters" in this already very long submission.

I appreciate the links to public code repositories.

---

## [Author Response]

Reviewer #1:This paper explores an important topic – how to test the efficacy of a bovine TB vaccine – using simulation models of TB transmission to test different trial designs. Overall, the analysis seem rigorous to the extent I could judge it, but ambiguity or missing details left me with several significant questions. The presentation of the paper is far from ideal overall – it is much too long, difficult to follow, and lack of required detail in the Materials and methods means it is not always clear how results have been derived. Really the paper needs major reorganisation, with a (much) shorter and selective main text, then a more detailed supplement to detail methods, parameterisation and sensitivity analyses.Issues:We have sharpened the focus of the paper, moving technical details of methods, models, review of transmission rates and sensitivity analyses to a new supplementary information document.- Subsection “Review of estimated cattle-to-cattle transmission rates for bovine TB” is critical – the failure of the AHVLA experiment fundamentally calls into question the feasibility of experimental studies of vaccine efficacy in the UK context. This really needs to be highlighted up front in the paper (not just in the Materials and methods). I note the authors contention that increasing group size and the proportion of the group initially infected would likely substantially increase infection rates, but given the cost of such studies and the need to demonstrate that the results will likely transfer to the field setting, I would suggest that funding any large-scale transmission study should be contingent upon achieving a higher attack rate in a small pilot (e.g. 50 animals, 25 infected, 25 uninfected). Such a pilot would also give invaluable data with which to better power future vaccine studies.

We acknowledge, and to a certain extent, share this concern about the feasibility of natural transmission studies using UK reactor animals. However, specific aspects of the design of the Khatri et al. study make interpretation of the findings challenging. Indeed reviewer #2 takes the contrasting view that our discussion of the AHVLA study was too negative as the rate of transmission seen in this study was not inconsistent with estimates from the field.

As discussed in the manuscript, and acknowledged by reviewer #1, the practicality of such studies depends on how transmission rates scale with group size. Likewise, the extent to which the rate of transmission seen in the Khatri et al.study is consistent with field estimates depends critically on what we consider the effective group size was in this study. We have estimated the effective (frequency dependent) transmission rate from the results of the AHVLA transmission experiment. The maximum likelihood estimate from this data is indeed comparable to the field estimate (SOR model) for a herd size of 10 animals which corresponds to the size of pens within the barn used at AHVLA. This grouping was imposed by the physical design of the facility and is likely to have reduced the opportunities for transmission between the 100 animals that were held within the same barn. However, perhaps the most surprising outcome of this study was that out of the eight transmission events observed two were associated with a genotype that was not present in reactor animals housed within the same pen. As transmission appears to have taken place between pens, the group size for this experiment could be argued to be the total number of animals in the barn. In this case, the maximum likelihood estimate of transmission would be far less consistent with field estimates, falling within the lower tail of the posterior distribution estimated from field data.

However, the posterior distribution for the transmission parameter estimated from the AHVLA study ranges over several orders of magnitude and includes both of these extreme assumptions for the effective group size. Given the level of uncertainty in this estimate we chose not to dwell on this issue in the original manuscript. We would further hesitate to rule out the feasibility of a transmission study in the UK solely on these data which were compromised by the practical constraints of the available facilities at AHVLA.

To address the concerns of both reviewer #1 and #2 we have added this analysis and comparison to the new supplementary information file. However, we concur that there is a strong argument to be made for a smaller scale pilot transmission study carried out and have added this to the Discussion. We would go further and suggest that carrying out such studies in endemic countries, although ruled out by EFSA for policy decisions in the UK/EU, would be a more practical way to establish the mode of action of BCG. We are currently planning such experiments in Ethiopia and India based on the design presented in this study and have added this wider biological context in the discussion of the revised manuscript.

- How exactly are the expected values plotted in Figure 3A (and like figures) being estimated? No details are given in the Materials and methods. I'm guessing it is the average of the estimated VE over a large number of simulations of the experimental transmission study of X herds? How was VE calculated – using the expressions given in Figure 1? Giving some representation of the 95% range of the estimates from single experiments would be useful.- Likewise, how exactly are the estimates of power calculated in Figure 3B? I presume from their simulations? In doing so, are they also simulating a 2-level analysis of the simulated trial results (i.e. accounting for variation between herds)? What primary end-point is being evaluated – a difference in attack rates between vaccinated and control herds, or estimation of VE to some level of precision (e.g. +/-0.05). I would suggest the latter is more useful. i.e. for a fixed set of assumptions about VE, run 1000 simulations of the trial in X herds, and count the number of simulations, Z, for which the desired measure of efficacy is estimated to within +/-0.05 (say) of its true value. Power is then Z/1000.

This was a key omission in the original submission and we appreciate the opportunity to correct this. In the reorganized manuscript, we have added a new Materials and methods section in the main manuscript which focuses on the definition of efficacy measures from field and experimental designs and the explicit calculations used to power each respective trial design.

On reflection, we realize that the distinction between how, and why, models have been used to inform field and experimental designs was not made clearly or powerfully enough in the original manuscript. As a consequence, one of our key messages – that field trials are a fundamentally imprecise way of assessing the mode of action of cattle vaccination with BCG – was also obscured.

Experimental transmission studies can be designed to directly estimate the effect of vaccination on susceptibility to infection (ϵS) and reduction in infectiousness (ϵI) through either a mechanistic (e.g. chain-binomial) or survival analysis. The assumed effect size and the expected effect size in this situation are the same and the power to estimate an effect depends only on the expected rate of transmission that can be achieved.

Use of a mechanistic model in a field trial setting to directly estimate εS and εI is not feasible in this setting due to our inability to quantify or control the extrinsic infectious pressure acting on cattle from sympatric wildlife reservoirs. Randomisation can be used to deal with this confounder, but the cost is we must use population level relative risk measures of efficacy which although related to the assumed individual level efficacy of vaccination (ϵS and ϵI) are population level measures whose expected value also depends on the rates of transmission within herds, the balance of intrinsic and extrinsic transmission rates and the frequency of testing and removal of reactor animals.

This was the motivation for using within-herd transmission models, estimated from field data and representative of the range of herd demographics and transmission settings seen in GB to predict the likely effect size of these relative risk measures of efficacy for different assumed individual effects (ϵS,ϵI). The variability in these measures from model simulations is extensive. In our original submission, we omitted confidence intervals on the plots of predicted effect size due to the difficulty in visualizing such wide and overlapping distributions. While this uncertainty is reflected in the low power to see a significant effect of vaccination for small numbers of herds, we acknowledge that this does not make clear the high probability of estimating a negative efficacy of vaccination for small effects. To bring home this point we have added supplemental plots that illustrate the full range of the predictive distribution for each relative risk measures of vaccine efficacy for the most optimistic individual level effect (Figure 3—figure supplement 1, Figure 3—figure supplement 4 and others in supplementary information).

While we agree with both reviewer #1 and #2 that precision is normally a more useful measure of the power of a trial design, in the face of such fundamental imprecision it strongly believe it is not meaningful here. The “low-bar” of estimating any significant effect of vaccination is, we would argue, the best we might hope to achieve from a field setting. The large risk that remains that an underpowered trial will estimate a (statistically significant) negative efficacy is our main argument that, in particular for quantifying the mode of action and impact of BCG on infectiousness, controlled natural challenge designs are the only realistic proposition.

- "Clearly the latter methods have a higher power and that is the method that is used for estimation and calculation of the post-hoc power from the simulation studies described below (A4)". I am confused. What does A4 refer to? Which tables and figures use the FS based approximate analytical calculation and which use direct simulation? Precise details of how vaccine efficacy and power was calculated from simulations of the experimental design should be given. I don't frankly see that the analytical approach in the subsection “Sample size calculations for natural transmission study” adds anything very much. Given any experiment must have a fixed duration, what is important is the net attack rate seen in the vaccinated and control animals over the duration of the experiment. This depends on the transmission rate (and how that varies as a function of the time from infection) and the duration of the study phases (i.e. contact time). Going from transmission rates to R and back confuses things, at least for me. I would rather see Figure 10 show the posterior distribution for transmission rate (including the herd size factor), therefore. This would be more informative than the estimates given in Table 12. It would also allow Figure 9 to be removed – which doesn't add anything informative beyond Figure 11 in my view – indeed the addition of the red vertical dashed lines to Figure 9 is confusing.

Apologies first about the inclusion of this reference to A4, a post-hoc simulation study carried to benchmark the analytical results for a range of simulation scenarios. In the interests of length and given the consistency and robustness of the analytic results we chose not to include this simulation study in this manuscript.

While we appreciate, the reviewers point about moving between transmission rates and reproduction ratios we disagree that transmission rates – even when scaled by herd size – are a more biologically meaningful quantity. In addition to potentially scaling with group size, estimated transmission rates depend on the form of the assumed latency distributions between infection and infectiousness associated with a particular model. The reproduction ratio for a particular duration of experiment folds in this additional information and is therefore a more meaningful comparator. Furthermore, the reproduction ratio serves as a design parameter (as noted by reviewer #2) for the existing experimental design theory that we depend on for the proposed design.

However, we agree that the presentation of the analytical results that are used to power the proposed natural transmission study could have been more clearly communicated. To this end we have generated new simplified figures that explore power as a function of group size and three discrete contact times of one, two and three years (Figure 5) and replace Figures 9, 11 and associated tables from the original manuscript.

- Continuing in the same vein, the paper gives the impression that Table 11 is driven by the results of Table 10, which misses the subtleties of Figure 11. This latter figure is the most interesting in the paper, but I didn't understand some of the trends in Figure 11. Why does increasing contact time decrease power (for fixed group size) for some model variants and vaccine efficacies? I can only assume this is because infection rates are saturating in both groups. However, if the experiment was analysed making use of all the DIVA test results in a survival analysis, this shouldn't matter – the higher infection hazard in the control group animals should still be resolvable in the first phase, and the lower infectiousness of the vaccinated animals in the second phase. As I've said above, the authors need to give precise details of how power is being estimated from the simulation for Figure 11 (see above) – how are the (simulated) experimental results being analysed, what is the trial primary end-point (i.e. what statistical test is being examined when calculating power)? Again, given the cost of such experiments, analysis needs to make best possible use of the data collected – which survival analysis is more likely to achieve than simple comparison of final attack rates.

Once again, apologies for the lack of detail in the specific power calculations and use of simulation (for field trial designs) and analytic results (for experimental designs). As the reviewer highlights the sample size calculation for the proposed experimental design is based on the final size distribution. Thus, based on the analytic calculation power will decrease with the difference in proportion infected within the vaccinated and control group – in particular when infection saturates in both groups. We agree that using the time series information provided by DIVA testing has the potential to greatly increase statistical power through the use of a survival or mechanistic chain-binomial transmission model. Indeed, we carried out such an analysis based upon the chain-binomial model in the simulation study mentioned above and found the estimates were both unbiased and that they validated the robustness of the analytic estimates of samples sizes as described in this paper.

We have updated the paper to make clear that these methods would be the most powerful way to analyse data from such experimental designs, but that the analytic final size method provides a robust, and most importantly conservative, method of sample size calculation.

- In the first paragraph of the subsection “Group size and duration of transmission studies under different transmission scenarios” – What are Figures B3 and B4 and what is Table B1? Assuming Table B1 is actually Table 12, how are the values given in that table used to generate Figure 9 on? In particular, were the first 2 rows of Table 12 used for any simulations, or just the Conlan et al. estimates? As mentioned elsewhere, I would drop the Conlan 2012 results – presumably they were superseded by the 2015 ones, and they give rather optimistic results for the trial contact times.- Table 11, 25% effect size – the bottom-most rows have a group size of 800, while this group size isn't mentioned in Table 10. Is this a typo? As commented below, I don't feel including all the 2012 model variants adds anything here.- The second paragraph of the subsection “Comparison of field estimates of cattle-to-cattle transmission rates” is unnecessary. Presumably the 2015 models are preferred, so reference to and results relating to the 2012 model (half of Figure 8, Figures 9, 11, Figure 11—figure supplements 2, 4, half of Table 11) can be removed.

Apologies once more for the formatting errors which were introduced during the submission process and the lack of clarity between numerical and analytic results. On advice, we have dropped the Conlan, 2012 results from the revised manuscript and simplified the presentation of sample size calculations as discussed above.

Reviewer #2:This paper is a well written, thorough and painstaking analysis of a narrow technical issue, sample size calculations for a hypothetical trial of vaccination to protect cattle from bovine TB. It is a substantial and technically useful piece of work, though not easily generalizable given the specific and complicated details of bovine TB epidemiology and management in the UK.I agree with the statement (subsection “Conceptual design to estimate vaccine efficacy and herd level effectiveness”, fifth paragraph) that it is important to evaluate the impact of vaccination on infectiousness as well as susceptibility – as a rule the former is ignored.The prediction that there would be, in the situation modelled, only a very small indirect impact of vaccination means that very large sample sizes would be needed for a trial to detect it. The situation modelled includes current test and slaughter practices, but presumably if a vaccine were to be used it would not be used in conjunction with these. If it was, as is spelt out later, the additional benefit would be very small and presumably not cost effective. Or is that the point the authors wish to make? Either way, a vaccination-only scenario would be of interest (regardless of current EC requirements).The way statistical power is estimated (more than "slightly" unusual in my view – Discussion, third paragraph) also makes the study less generalizable. It would help to set out what efficacy we are looking for, greater than zero seems a very low bar. (What's more, the subsequent discussion about the DIVA test implies that even if the vaccine itself had zero efficacy there would be some effect of the more sensitive test – Discussion, sixth paragraph). These impacts could be partitioned by varying parameter settings appropriately.The role of extrinsic infection (Discussion, eighth paragraph) could also be explored more systematically. The issue of testing efficacy locally when the ultimate aim would be to intervene over a whole population is problematic for many intervention trials for infectious disease.Overall, I felt that, though a rather daunting volume of results are presented already, more could have been done to dissect out the likely multiple contributors to the low efficacy anticipated in a herd level vaccination trial.

The European Union and UK policy makers have consistently held the view that the use of cattle vaccination will only be acceptable as a supplement to ongoing test-and-slaughter. We agree with the reviewer that this poses a particular barrier for the potential cost-effectiveness of any roll out as well as limiting the likely success of any prospective field trials. As such, we also agree that it is important to consider the potential for a vaccination-only trial and have addressed this question in the revised manuscript.

Rather than replicating our analysis of the full range of efficacy and persistence measures for this new scenario, and to address the related question as to the factors contributing to the low expected efficacy, we examine the impact that retention of DIVA test positive animals has on the predicted number of reactor animals seen in different trial scenarios.

In Figure 4 of the revised manuscript we present posterior predictive distributions for the within-herd prevalence of bTB. These distributions reveal a high frequency mode of single (or very few) reactor TB incidents. This prediction is consistent with the distribution of reactor animals seen within UK herds where less than half of bTB breakdowns have more than 1 reactor animal disclosed. This empirical observation is also reflected in the cattle-to-cattle reproduction ratios from within-herd models which suggest a bi-modal distribution of sub-critical (R_0_ < 1) and super-critical herds (R_0_ > 1).

The low predicted attack rate in the majority of trial herds leads to relative risk measures systematically underestimating vaccine efficacy. Indeed the ability to discriminate between the attack rate in vaccinated and unvaccinated groups rests almost entirely with the relatively few herds which experience a high attack rate. This heterogeneity is, we would argue, the fundamental constraint that makes relative risk measures at the population level a poor way to assess the efficacy of cattle vaccination for bTB in Great Britain.

The new Figure 4 illustrates the extent to which retaining reactor animals or increasing the duration of the trial might increase the expected attack rate. Perhaps surprisingly, retaining reactor animals has almost no effect on the expected herd-level prevalence for a three-year trial. This is due to the long generation time of bTB transmission, requiring in excess of 15 years for model herds to reach an endemic equilibrium. Increasing the duration of trial has a more pronounced effect, but has little impact on the mode of low prevalence herds serving to thicken the long tail of herds which see a relatively higher rate of transmission.

The only feasible solution we can see to increasing the utility of relative risk measures of vaccine efficacy would be to target herds likely to see a higher rate of transmission. For example, if we could target large herds with a low turnover of animals we could increase the chances that a randomly selected herd has a reproduction ratio greater than 1. However, this would not be a practical design for a field trial due to the relative rarity of such farms (larger farms tend also to have higher turnovers) and the dependence on farmers to volunteer for participation. Even within model simulations, such targeting would only have limited effect due to the high level of stochasticity and parametric uncertainty. Likewise, targeting herds with a high extrinsic rate of transmission could increase the probability of exposure during a trial but is impractical due to the complete lack of data or methodology to quantify the local infection risk for particular herds.

Practicality notwithstanding, such targeting of herds would not satisfy the EU/EFSA requirement that field trials be carried out in herds that are representative of European production conditions.

The way statistical power is estimated (more than "slightly" unusual in my view – line 417) also makes the study less generalizable. It would help to set out what efficacy we are looking for, greater than zero seems a very low bar. (What's more, the subsequent discussion about the DIVA test implies that even if the vaccine itself had zero efficacy there would be some effect of the more sensitive test – lines 453-5). These impacts could be partitioned by varying parameter settings appropriately.

As argued above in our response to reviewer #1 one of our main conclusions is that the imprecision of relative risk measures of vaccine efficacy for bovine TB would generate a high risk of estimating a negative vaccine efficacy even when a true effect exists. In this context an efficacy greater than zero may in fact be too high a bar for any vaccine, a point we now make more explicit in the revised paper.

Reviewer #3:General Comments/Suggestions:This manuscript describes the adaptation of previously published models to understand how field trials might (or might not!) detect the benefits of a bTB vaccine deployed in Britain. In general, the methods and conclusions seem sound, and represent an important warning on relying on these sorts of trials. I did have some problems understanding the interpretation results shown in the figures associated with this manuscript, and I suspect that some figures may not be referred to correctly. I have included some suggestions below on how to make the figures more easily readable. The mathematical typesetting in the manuscript also made it somewhat less readable – typesetting that sets apart mathematical entities like R and R_0_ more clearly would have helped me, and would also have limited confusion when "R" is used both as a reproduction parameter, and as a compartment (e.g. in the SORI model).Typesetting of all mathematical entities using equation editor has been carried out in the revised manuscript.Because this work is based on mathematical modelling, I would urge the authors to make as much of the modelling code as possible available on a public repository, or provide links to the previously-published code used. Publishing code in this way makes the work much more reproducible.

A data repository containing all simulation code, parameter sets, data and script to reproduce every figure in the manuscript has been made available with the resubmission.

[Editors’ note: what follows is the authors’ plan to address the revisions.]

Reviewer #1:The rewritten paper is much clearer and more comprehensible. My few technical comments are detailed below. I do have more major issues with the conclusions and tone of the paper however:- Given the problematic experience with previous transmission experiments, my own conclusion from reading this paper was that relatively small field trial of 200 herds would give valuable information about the likely veterinary health impact of vaccination at the individual animal level (but that a trial with 500 herds would be needed to measure herd-level effectiveness).- Data from an experimental study is fundamentally different in quality than that from a randomised field trial. In human public health, the former might be viewed as equivalent to a pre-phase II human challenge study, which gives proof of concept. It does not guarantee that the results can be read across to the natural setting – which is why phase III trials are still needed.- The authors seem unnecessarily pessimistic about what their simulations imply for the feasibility of field trials, at least at the individual animal level. I interpret Figure 3 as showing that a trial run in 200 herds would have excellent power at measuring the direct effect of vaccination, and reasonable power at measuring the 'total' effect.- Yes, measuring indirect effects is difficult, but arguably is addressable in a post-marketing (phase IV) implementation study.- I think there are issues with a cluster-randomised trial with 50% coverage in each cluster (herd). Even for the vaccinated animals, outcomes will be different in a herd with 100% vs 50% coverage of a leaky vaccine. Plus, presumably the goal for any widescale vaccination policy would be 100% coverage? A 3-arm trial with herd level vaccination coverage of 0, 50% and 100% in the three arms might be more informative. Comparing attack rates in the 0 and 100% arms would give a measure of total effect (the most important outcome). Comparing vaccinated animals in the 50% and 100% coverage arms and unvaccinated animals in the 50% and 0% arms would give more information (and therefore power) to differentiated impacts of vaccination on infectiousness and susceptibility.- Appendix 2 on whole-herd effectiveness is interesting and critically important (to the extent I would much rather see this in the main text and the discussion of experimental transmission studies in an appendix) – and in my view calls into question the whole viability of vaccination, if the overall impact on herd breakdowns is really only likely to be in the range of 10-20%. Putting that (major) issue to one side, the results in this section also highlight the potential benefits of a 3-arm design.- Regarding the discussion of bias in RR measures – it is unsurprising that such measures underestimate ε_s_ – it's the difference between comparing a hazard with a cumulative hazard.- Indeed, Figure 3 (and the supplementary version) seems to show that one could use models to quite reliably go back from the measured relative risks to the underlying effect of vaccine on susceptibility – albeit not on infectiousness.Reviewer #2:The first round of reviews seems to have picked up a large number of errors and presentational issues. The authors have addressed these fairly comprehensively and the manuscript is greatly improved as a result. If the topic is thought appropriate for eLife then I recommend that the manuscript is now acceptable for publication.Reviewer #3:In general I am satisfied with the re-organisation and changes made. I find the manuscript is more focused and easier to get through in its current form. I still find the Appendices somewhat arduous, and would encourage the authors to consider any last-minute changes they can make to streamline them, but I accept that sometimes Appendices with technical content can be long.I was a bit disappointed that the authors felt they could not address the impact of the distribution of latencies on the design, but accept their justification that, with the very high level of uncertainty on these distributions, they do not want to "muddy the waters" in this already very long submission.I appreciate the links to public code repositories.

We thank the reviewers again for the careful review of our manuscript and insightful comments.

In particular the reviewing editor makes careful reference to the standards and practice of field evaluation of vaccines in human public health to make a case for the inherent usefulness of a field trial of BCG in cattle. We find very little to disagree with these comments and acknowledge that the presentation of our results may have obscured our support for the proper evaluation of BCG vaccination in the field.

We whole-heartedly agree that field evaluation of the individual efficacy of BCG in 200 herds would be feasible and provide valuable information to inform decision on deployment. Indeed as part of the government funded Triveritas consortium, authors on this paper made this specific recommendation to Defra. We agree that a 3-arm design would additionally provide actionable information on the effectiveness of BCG under conditions of deployment in the field. Such a design was not offered within the Triveritas report due to concerns over the likely cost of field trials and the requirement to mitigate the risk of failure. Hence, a two-stage trial was proposed with progression to a larger trial to assess effectiveness contingent on the success of a smaller scale trial to demonstrate individual efficacy.

However, the intractable – not impossible – challenge of field evaluation of BCG in Great Britain comes from the relationship between the scale of such trials, the current legal status of vaccination and the expectation of what can be achieved by a single trial. When Defra commissioned the design of field trials and evaluated the proposed designs they took the view that field trials must satisfy all of the recommendations of the EFSA report including the two key requirements highlighted in our manuscript relating to indirect transmission and use of vaccination as a supplement to testing rather than a replacement. This position goes beyond the EFSA recommendations themselves which clearly state that deviation from the recommendations could be made provided there was a strong scientific reason for the decision.

A key objective of this paper is to provide a clear argument for the value of carrying out a series of more tractable trials which are more each more appropriate for assessing different facets of the efficacy of BCG. The analogy to the phases of vaccine trials raised by the reviewing editor is a powerful one, which we have used ourselves in private discussions with Defra. In particular we agree with the reviewing editor that addressing the indirect effects of vaccination would potentially be achievable in a phase IV trial after deployment. However, the current legal situation is that the use of cattle vaccination is explicitly prohibited by EU, EC and GB law. For the purposes of a trial BCG could be used under authorisation of the Secretary of State – however those animals would likely be subject to movement and trade restrictions for the rest of their life. Deployment of vaccination at scale would require a change in UK and EU/EC law (subject still to negotiations surrounding trade agreements following Brexit) which at the moment is linked to satisfying all of the requirements of the EFSA report including the indirect effects of vaccination. The UK government’s interpretation of the EU position is that the impact of BCG on transmission of bTB must be demonstrated before marketing authorisation is given to allow deployment of the vaccine at scale. We believe that this is an unreasonable expectation. Although we agree that future field evaluation of indirect effects would be important, we also contend that experimental quantification of the mode of action of BCG would be a more tractable solution to break this legal and political impasse.

We agree with the reviewing editor about the importance of the Appendix on whole-herd effectiveness. However, we would disagree that these results call into question the viability of vaccination per se, rather that they bring into question the viability of vaccination that is only used as a supplement to ongoing test-and-slaughter. The official Defra position is that field trials have been postponed until the diagnostic DIVA test – on which the power of our trial designs critically depend – is appropriately validated. However, a contributing factor to this decision was the overwhelming challenge of seeing a positive cost-benefit for vaccination after factoring in the additional costs of a new diagnostic test and the vaccine itself. We believe that vaccination could still play an important role in the control of bovine tuberculosis – particularly in developing countries where test-and-slaughter is not acceptable economically, or in the case of India, ethically. In the UK we would also argue that the use of vaccination as a replacement for test-and-slaughter could transform the case for vaccination – and the potential to evaluate its effectiveness robustly in the field.

To address these major issues from the reviewing editor we propose the following plan of action:

- Swap the vaccine effectiveness scenarios into the main body of the paper and move the experimental design to a technical appendix

- Revise the discussion to:

Explicitly state that field evaluation of BCG is viable, providing it focuses on validating the direct individual level efficacy

Explicitly state that a trial becomes considerably more challenging if:

Indirect effects of cattle vaccination must be estimated before deployment of the vaccine

Vaccination is only considered as a supplement to test-and-slaughter

Propose that these challenges can be mitigated by planning a phased series of trials in line with established standards in human public health:

Use natural transmission experiments to address the mode of action of BCG before field evaluation (Phase II)

Field evaluation of individual efficacy of BCG in 200 herds (Phase III)

Trials to establish the effectiveness and indirect effects of BCG (Phase IV)

Caution that the use of vaccination only as a supplement to test-and-slaughter will necessarily limit the perceived benefits of vaccination to farmers and argue for policy makers to consider strategies for the use of vaccination as a replacement for test-and-slaughter.

We also propose to make the following more minor changes in response to the reviewing editors comments:

1) “Regarding the discussion of bias in RR measures – it is unsurprising that such measures underestimate ε_s_ – it's the difference between comparing a hazard with a cumulative hazard.”

We will add this clarification, but feel the general exposition of between-herd variability is important and should be retained.

Timetable for revisions

Based on the action plan outlined above we would expect to be able to prepare a revised manuscript within two-weeks. Recalculating the power curves using a non-parametric Mann-Witney test would require an additional two weeks to edit, check and rerun the code to update the eight affected figures in the main manuscript and technical appendix. As argued above, we believe that the power calculations based on relative risk are sufficient and appropriate to support our conclusions, but are happy to make this revision should the reviewing editor require it.